# GOLD: Graph Out-of-Distribution Detection via Implicit Adversarial Latent Generation

**Danny Wang, Ruihong Qiu, Guangdong Bai, Zi Huang**
The University of Queensland
{danny.wang,r.qiu,g.bai,helen.huang}@uq.edu.au

## Abstract

Despite graph neural networks' (GNNs) great success in modelling graph-structured data, out-of-distribution (OOD) test instances still pose a great challenge for current GNNs. One of the most effective techniques to detect OOD nodes is to expose the detector model with an additional OOD node-set, yet the extra OOD instances are often difficult to obtain in practice. Recent methods for image data address this problem using OOD data synthesis, typically relying on pre-trained generative models like Stable Diffusion. However, these approaches require vast amounts of additional data, as well as one-for-all pre-trained generative models, which are not available for graph data. Therefore, we propose the GOLD framework for graph OOD detection, an implicit adversarial learning pipeline with synthetic OOD exposure without pre-trained models. The implicit adversarial training process employs a novel alternating optimisation framework by training: (1) a latent generative model to regularly imitate the in-distribution (ID) embeddings from an evolving GNN, and (2) a GNN encoder and an OOD detector to accurately classify ID data while increasing the energy divergence between the ID embeddings and the generative model's synthetic embeddings. This novel approach implicitly transforms the synthetic embeddings into pseudo-OOD instances relative to the ID data, effectively simulating exposure to OOD scenarios without auxiliary data. Extensive OOD detection experiments are conducted on five benchmark graph datasets, verifying the superior performance of GOLD without using real OOD data compared with the state-of-the-art OOD exposure and non-exposure baselines. [1]

## 1 Introduction

The proliferation of Graph Neural Networks (GNNs) across diverse domains and real-world applications has underscored the importance of robust and reliable predictive systems (Kipf & Welling, 2017; Hamilton et al., 2017). Their performance relies crucially on the assumption that the testing data follows the same distribution as the training data (Li et al., 2023; Kipf & Welling, 2017; Yu et al., 2023; Hamilton et al., 2017). This assumption is frequently violated in practice, as real-world graph data is generally filled with out-of-distribution (OOD) instances (Bitterwolf et al., 2020; Chen et al., 2022; Ding et al., 2021; Zhou et al., 2022; Li et al., 2022a; Yang et al., 2022). Consequently, inaccurate predictions will inevitably be made by the deployed models, which can be detrimental in critical areas like medical diagnosis and drug discovery (Cao et al., 2020; Ahmedt-Aristizabal et al., 2021; Giuffrè & Shung, 2023; Lee et al., 2023b; Ji et al., 2022). Thus, it is necessary to develop OOD detection methods to identify out-of-distribution instances that deviate from the training distribution (Yang et al., 2021; Ren et al., 2019; Lang et al., 2023; Bazhenov et al., 2022).

Recent work has made significant strides in developing OOD detection techniques tailored for graph-structured data, primarily in three categories (Song & Wang, 2022; Huang et al., 2022a; Wu et al., 2023b; Stadler et al., 2021). (1) General OOD detection methods train the detector only with in-distribution (ID) data from the training set (Ding & Shi, 2023; Ma et al., 2023; Liu et al., 2023b; Wang et al., 2024). This process involves fine-tuning a classifier and learning graph representations

---

[1] Code is available at https://github.com/DannyW618/GOLD.

to improve the model's OOD detection performance using various scoring metrics. (2) A more effective method for OOD detection is OOD exposure, which takes advantage of exposing the detector with additional OOD samples during training (Wang & Li, 2023; Wu et al., 2023b; Hendrycks et al., 2019; Koo et al., 2024). These methods generally require an extra dataset containing OOD samples and the detector is trained to discriminate the ID training data with these OOD data. (3) More recently, OOD synthesis methods have been proposed for image data, mainly leveraging pre-trained generative models, e.g., Stable Diffusion (Rombach et al., 2022), to create OOD samples that lie on the boundary of ID data (Tao et al., 2023; Du et al., 2023; Wu et al., 2023a; Zheng et al., 2023).

Despite the effectiveness of OOD exposure-based methods over general OOD detection methods, two challenges remain: (1) For the OOD exposure approaches using a real and additional OOD dataset, acquiring these extra OOD samples is often infeasible during model training in the real world. Furthermore, relying on the additional OOD dataset to guide the detector in distinguishing the ID and OOD data could lead to an inaccurate decision boundary. This is because the training logic assumes that the exposed OOD data can represent the distribution of OOD data from test scenarios, which has no guarantee in real-world (Du et al., 2022; Vernekar et al., 2019). (2) Although OOD synthesis-based approaches have been proposed to resolve the lack of unknown data, these methods typically rely on pre-trained models built upon substantial amounts of auxiliary data (Huang et al., 2022c; Tao et al., 2023; Du et al., 2023; Shen et al., 2024). Moreover, the lack of a one-for-all pre-trained generative model for graph data hinders the synthesis of OOD data using simple plug-and-play models (Liu et al., 2023a). Thus, this presents the key motivation:

***How to enhance graph OOD detection by exposing to OOD scenarios without auxiliary data?***

In light of the above challenges, the intuition of this work is to generate and expose pseudo-OOD samples solely based on the ID training data to ensure effective OOD detection. To achieve this, we propose an implicit adversarial training framework with a novel alternating optimisation schema by training: (1) a latent generative model (LGM) to **regularly generate embeddings similar to the in-distribution (ID) embeddings from an evolving GNN**, and (2) a GNN encoder and an OOD detector to accurately classify ID data while **increasing the energy divergence between these generated embeddings and the ID embeddings**. This novel approach implicitly transforms synthetic embeddings into pseudo-OOD instances relative to the ID data, effectively simulating OOD exposure without auxiliary data. Evident in Figure 1, the initially similar energy distributions after LGM training diverge post-training, which implicitly separates the embedding dis-

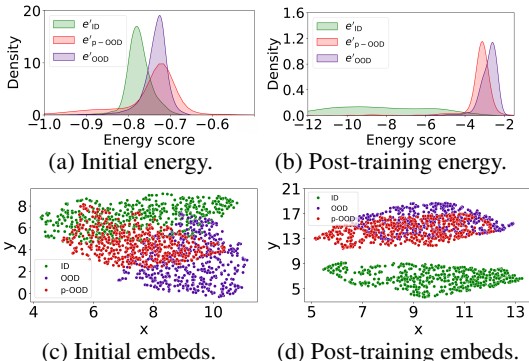

(a) Initial energy.    (b) Post-training energy.

(c) Initial embeds.    (d) Post-training embeds.

Figure 1: Motivation of GOLD: The initially close energy distributions (a) after training the latent generative model, become separated after training GOLD (b), where the initial pseudo-OOD (p-OOD) embeddings (embeds.,) (c) implicitly diverges from the ID data and resembles real OOD instances (d).

tributions, ensuring the pseudo-OOD data resemble close to the real OOD instances. The main contributions of this paper are summarised as follows:

- We propose GOLD, a novel non-OOD exposed synthesis-based framework for graph OOD detection. GOLD includes a unique implicit adversarial training paradigm for effective pseudo-OOD synthesis, which is achieved by a latent generative model and a novel detector.
- We conducted extensive experiments on five benchmark datasets. Without auxiliary OOD data, GOLD achieves state-of-the-art performance compared with non-OOD and OOD exposure methods, with the best improvement of FPR95 reduced from 33.57% to 1.78%.

## 2 PRELIMINARY

Generally, for a node classification problem, a graph is denoted as $\mathcal{G} = (\mathbf{X}, \mathbf{A})$, where $\mathbf{X} \in \mathbb{R}^{n \times d}$ is the node feature matrix with $n$ nodes and feature dimension $d$, and $\mathbf{A} \in \mathbb{R}^{n \times n}$ is an adjacency

matrix indicating the connection among nodes. Each node is associated with a label $y \in \{1, 2, ..., C\}$ indicating a total of $C$ classes. For out-of-distribution detection, there are generally two main tasks:

**Task I: In-distribution classification.** To formulate the node classification problem for in-distribution data, given test nodes from the same distribution as training nodes, $P_{train}(\mathbf{X}, \mathbf{A}) = P_{test}(\mathbf{X}, \mathbf{A})$ and the conditional distribution $P_{train}(\mathbf{y}|\mathbf{X}, \mathbf{A}) = P_{test}(\mathbf{y}|\mathbf{X}, \mathbf{A})$, the task is to develop an $L$-layer GNN classifier to predict the label $\mathbf{y} \in \mathbb{R}^n$ for the testing nodes with trainable parameters in the GNN classifier (see Appendix A.5 for details of GNN):

$$\mathbf{y} = \text{Softmax}(\text{GNN}(\mathbf{X}, \mathbf{A})). \tag{1}$$

**Task II: Out-of-distribution detection.** To detect the testing nodes coming from a different distribution from the training data, where $P_{train}(\mathbf{X}, \mathbf{A}) \neq P_{test}(\mathbf{X}, \mathbf{A})$ and the conditional distribution $P_{train}(\mathbf{y}|\mathbf{X}, \mathbf{A}) \neq P_{test}(\mathbf{y}|\mathbf{X}, \mathbf{A})$, the task is to require an OOD detector $F$ to output a binary prediction for the testing nodes. $F$ is usually built upon the output from the classifier GNN with $F(\mathbf{x}, \mathbf{A}; \text{GNN}) = 1$ for data from in-distribution and $F(\mathbf{x}, \mathbf{A}; \text{GNN}) = 0$ for data from out-of-distribution (Wu et al., 2023b; Liu et al., 2020; Yang et al., 2023a).

**Energy Score-based Detector.** Recent work indicated that using the energy score from logits in the classifier can benefit OOD detection (Liu et al., 2020; Wu et al., 2023b; Grathwohl et al., 2020). The energy score $e$ for a node $i$ is defined as:

$$e_i = -\log \sum_{c=0}^{C-1} \exp(\mathbf{z}_{i,c}), \tag{2}$$

where $e_i \in \mathbb{R}$ is the energy score of node $i$, $\mathbf{z}_i \in \mathbb{R}^C$ is the logits for node $i$ output from the classifier $\mathbf{Z} = \text{GNN}(\mathbf{X}, \mathbf{A}) \in \mathbb{R}^{n \times C}$, and $c$ is to select the logit of the $c$-th element of $\mathbf{z}_i$. Therefore, the energy score-based OOD detector for a node $i$ is instantiated with a threshold $\tau$ as:

$$F(\mathbf{x}_i, \mathbf{A}; \text{GNN}) = \begin{cases} 0, & \text{if} \quad e_i \geq \tau, \\ 1, & \text{if} \quad e_i < \tau. \end{cases} \tag{3}$$

The training of this energy score-based OOD detector is generally based on an energy regulariser (Liu et al., 2020), which maximises the difference between the energy scores from in-distribution data ($P_{\text{ID}}$) and out-of-distribution data ($P_{\text{OOD}}$) with two scalar thresholds, $t_{\text{ID}}$ and $t_{\text{OOD}}$:

$$\max_{\text{GNN}} \mathcal{L}_{\text{EReg}}, \text{ where } \mathcal{L}_{\text{EReg}} = \mathbb{E}_{i \sim P_{\text{ID}}} \left[ \max(0, t_{\text{ID}} - e_i) \right]^2 + \mathbb{E}_{j \sim P_{\text{OOD}}} \left[ \max(0, e_j - t_{\text{OOD}}) \right]^2. \tag{4}$$

**Energy Propagation for OOD Detector.** To facilitate the energy score for graph data, GNNSAFE (Wu et al., 2023b) proposes an energy propagation schema that emulates label propagation for effective OOD detection. This propagated energy is then fed into the objective in Eq. 4:

$$\mathbf{e}^{(k)} = \alpha \mathbf{e}^{(k-1)} + (1 - \alpha)\mathbf{D}^{-1}\mathbf{A}\mathbf{e}^{(k-1)}, \tag{5}$$

where $\mathbf{e}^{(k)} \in \mathbb{R}^{n \times 1}$ is the energy scores for $n$ nodes after $k$-th energy propagation with $\alpha \in [0, 1]$ controlling the concentration of energy. $\mathbf{D}$ is the degree matrix of graph $\mathcal{G}$. In the following, the energy scores in our framework will be the propagated energy scores and will be used interchangeably.

## 3 GOLD

In this section, the GOLD framework for graph OOD detection is described with illustration in Figure 2. In summary, GOLD is trained with a novel implicit adversarial objective that optimises a latent generative model (LGM), a GNN classifier, and an OOD detector. The LGM aims to generate embeddings akin to ID data, while the implicit adversarial objective encourages divergence between the ID and OOD energy scores derived from GNN and detector. This process implicitly transforms the synthetic embeddings into pseudo-OOD, effectively facilitating synthetic OOD exposure.

The GNN classifier is trained to maximise the log probability of the ground truth classification label:

$$\max_{\text{GNN}} \mathcal{L}_{\text{CLS}}, \text{ where } \mathcal{L}_{\text{CLS}} = \log p(y|\mathbf{x}, \mathcal{G}_{\mathbf{x}}). \tag{6}$$

In the following, the detector and the latent generator are both built upon this GNN in GOLD.

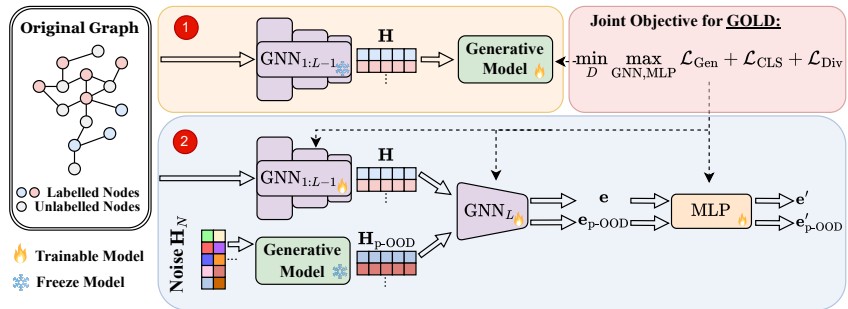

Figure 2: Overview of GOLD. Given an input graph, GOLD consists of two components: **Step ❶** trains a latent generative model using hidden representation $\mathbf{H}$ from a frozen GNN. **Step ❷** trains a GNN classifier and an OOD detector based on the ID data $\mathbf{H}$ and the latent generator generated pseudo data $\mathbf{H}_{\text{p-OOD}}$. The overall training is in an adversarial manner.

## 3.1 LATENT GENERATIVE MODEL AS PSEUDO-OOD GENERATOR

In light of the key motivation of exposing the model to OOD scenarios with generated data, an LGM is employed for pseudo-OOD synthesis. The model would take input from the encoded node representations $\mathbf{H} \in \mathbb{R}^{n \times d'}$, where $d'$ is the hidden dimension, of the GNN module after the $(L-1)$-th layer, which captures both the global and local information (Kipf & Welling, 2017):

$$\mathbf{H} = \text{GNN}_{1:L-1}(\mathbf{X}, \mathbf{A}), \quad \mathbf{Z} = \text{GNN}_L(\mathbf{H}, \mathbf{A}) = \text{GNN}(\mathbf{X}, \mathbf{A}). \quad (7)$$

The LGM aims to mimic and generate latent embeddings close to the ID representations. This is typically achieved by minimising a reconstruction loss or distance between a target and predicted embedding, i.e., if a latent diffusion model (LDM) (Ho et al., 2020) or a variational autoencoder (VAE) (Kingma & Welling, 2014) is used as the pseudo-OOD generator, the objective is given by:

$$\min_D \mathcal{L}_{\text{Gen}}, \quad (8)$$

$$\text{where } \mathcal{L}_{\text{Gen}} = \begin{cases} \mathbb{E}_{\mathbf{h}_0, \boldsymbol{\epsilon}, t}\left[\left\|\boldsymbol{\epsilon} - D\left(\sqrt{\bar{a}_t}\mathbf{h}_0 + \sqrt{(1-\bar{a}_t)}\boldsymbol{\epsilon}, t\right)\right\|_2^2\right], & \text{if LDM} \\ \mathbb{E}_{q_D(\mathbf{h}_{\text{p-OOD}}|\mathbf{h})}\left[\log p(\mathbf{h}|\mathbf{h}_{\text{p-OOD}})\right] - \text{KL}\left[q_D(\mathbf{h}_{\text{p-OOD}}|\mathbf{h})\|p(\mathbf{h}_{\text{p-OOD}})\right], & \text{if VAE} \end{cases}$$

with latent vectors $\mathbf{h}$ and decoder $D$. For LDM, $D$ is a denoising network that predicts and progressively removes noise during the backward denoising step. Contrary, VAE minimises a reconstruction and embedding distance loss, based on the input latent embeddings and the embeddings generated by the decoder $D$. The pseudo-OOD latent embeddings $\mathbf{h}_{\text{p-OOD}}$ can thus be generated using the decoder network $D$ with noise vector sampled from a normal distribution $\mathbf{h}_N \sim \mathcal{N}(0, \mathbf{I})$ via:

$$\mathbf{h}_{\text{p-OOD}} \sim P_{\text{p-OOD}}, \text{ and } P_{\text{p-OOD}} = P_D(\mathbf{h}_{\text{p-OOD}}|\mathbf{h}_N) \quad (9)$$

For comparison, VAE presents competitive performance and faster training time due to model design, while LDM remains efficient and performs better among (non-) OOD exposure methods when used in GOLD. Moreover, GOLD achieves the same inference time as SOTA baselines with any LGMs. This is because the generative model is not involved during inference. A detailed description of the two generative methods and their corresponding objective is provided in Appendix A.11, and further results about effectiveness and efficiency will be discussed in Section 4.1 and 4.6.

Note that at this stage, the synthetic embeddings generated by the LGM still imitate the ID data. In the following subsections, with a novel detector, an implicit adversarial training process will be introduced, which separates the synthetic embeddings from the ID representations, transforming it into pseudo-OOD instances.

## 3.2 OOD DETECTOR FOR ID AND PSEURO-OOD SEPARATION

Given that a trained latent generator can synthesise latent representations akin to the ID embeddings, the OOD detector is designed to pull apart the energy scores of ID instances from those of the generated data. This ensures a clear separation between the distributions, and through gradient flow to the

trainable GNN encoder, it implicitly separates the synthetic embeddings from the ID embeddings. Hence, the synthetic data is effectively transformed into pseudo-OOD instances relative to ID data, as shown in Figure 1, allowing the model to be exposed to OOD scenarios without the need for real OOD data. For clarity, in the following, the OOD exposure in previous methods will be replaced with the pseudo-OOD data generated by the LGM from Eq. 9.

In the general design of an energy-based OOD detector as in Eq. 2, the energy score is a combination of the prediction logits. To overcome the potential difficulties when the number of classes increases or when certain classes are unable to be accurately distinguished by the model, an MLP is applied to the energy and trained with an uncertainty loss as in Du et al. (2022) with $\phi$ as the softmax function:

$$\max_{\text{GNN, MLP}} \mathcal{L}_{\text{Unc}}, \text{ where } \mathcal{L}_{\text{Unc}} = \mathbb{E}_{i \sim P_{\text{ID}}} \log[\phi(\text{MLP}(e_i))_{[0]}] + \mathbb{E}_{j \sim P_{\text{p-OOD}}} \log[\phi(\text{MLP}(e_j))_{[1]}]. \quad (10)$$

The subscripts indicate the label of the corresponding logit value from the MLP model after applying $\phi$ (i.e., [0] represents the ID Class 0 and [1] represents the OOD Class 1). In addition to using this uncertainty objective, we aim to further transform the energy with the classifier output to enhance the separability of the energy:

$$e'_i = -\log[e^{\text{MLP}(e_i)_{[0]}} + e^{\text{MLP}(e_i)_{[1]}}]. \quad (11)$$

With the transformed energy $\mathbf{e}'$, we propose a new divergence regularisation to obtain a more diverged energy score distribution than the pre-transformed energy $e$ from the GNN classifier:

$$\max_{\text{GNN, MLP}} \mathcal{L}_{\text{DReg}}, \text{ where } \mathcal{L}_{\text{DReg}} = \mathbb{E}_{i \sim P_{\text{ID}}} \max\left(0, e_i - e'_i\right)^2 + \mathbb{E}_{j \sim P_{\text{p-OOD}}} \max\left(0, e'_j - e_j\right)^2. \quad (12)$$

We next show that the combination of the two proposed losses with pseudo-OOD data could enable the detector to produce more distinctive energy scores between distributions to assist OOD detection.

> **Proposition 1.** *The gradient descent on $\mathcal{L}_{Unc}$ and $\mathcal{L}_{DReg}$ will overall decrease (increase) the transformed energy $\mathbf{e}'$ for in-distribution (pseudo-out-of-distribution) instance, bounded by the given initial energy $\mathbf{e} \sim P_{ID}$ ($P_{p\text{-}OOD}$), respectively, for the detector MLP model.*

The proof is provided in Appendix A.1. Intuitively, the $\mathcal{L}_{\text{Unc}}$ aims to train the detector to classify the ID and OOD data with high probability under binary classification, which ensures the separability of embeddings. While the $\mathcal{L}_{\text{DReg}}$ aims to diverge the energy of ID and OOD based on the logits from this same detector. Therefore, the logits of ID data are expected to have a larger scale than the logits of OOD data, which leads to the energy score based on the logits for this binary classification providing a greater discrepancy between ID and OOD data. The empirical visualisation is shown in Figure 5 of Appendix A.2.

***Energy Divergence Objective***: Replacing $P_{\text{OOD}}$ with $P_{\text{p-OOD}}$, additionally with the $\mathcal{L}_{\text{EReg}}$ from Eq. 4, the objective to diverge the energy for the OOD detector is a combination with weight $\mu, \lambda, \gamma \in \mathbb{R}$:

$$\max_{\text{GNN, MLP}} \mathcal{L}_{\text{Div}}, \text{ where } \mathcal{L}_{\text{Div}} = \mu \mathcal{L}_{\text{EReg}} + \lambda \mathcal{L}_{\text{Unc}} + \gamma \mathcal{L}_{\text{DReg}}. \quad (13)$$

After optimising the MLP detector and GNN classifier with the final energy divergence objective $\mathcal{L}_{\text{Div}}$, the embeddings generated by the fixed LGM will implicitly diverge from the ID embeddings produced by the updated classifier. This divergence occurs because the energy scores generated by the detector are separated by the optimised objective, which will further train the GNN classifier via gradient flow. As a result, the LGM would effectively function as a pseudo-OOD generator.

## 3.3 IMPLICIT ADVERSARIAL OBJECTIVE

To accomplish the ID classification and the OOD detection, the overall objective of the pseudo-OOD synthesis and OOD detector can be formulated in an adversarial style, by combining Eq. 8, 6 and 13:

$$\min_{D} \max_{\text{GNN, MLP}} \mathcal{L}_{\text{Gen}} + \mathcal{L}_{\text{CLS}} + \mathcal{L}_{\text{Div}} \quad (14)$$

The intuition of this adversarial objective stems from the contradictory optimisation purpose from the individual objectives. When fixing the GNN encoder, $\mathcal{L}_{\text{Gen}}$ aims to optimise the LGM to minimise the gap between the generated pseudo-OOD embeddings and ID embeddings. This ensures the LGM can generate meaningful representations that are initially close to ID data, instead of generating meaningless and far away pseudo-OOD data. When fixing the generator, $\mathcal{L}_{\text{CLS}}$ and $\mathcal{L}_{\text{Div}}$

aims to optimise the GNN encoder and the MLP detector to maximise the discrepancy in the energy score between the pseudo-OOD data and the ID data, while keeping the GNN encoding for ID data meaningful for classification. Notably, GOLD does not directly generate pseudo-embeddings from the LGM against the ID embeddings but instead, the encoder and the detector implicitly pull the ID embeddings away from the generated pseudo-OOD embeddings via the energy score divergence.

We note that adversarial training schemas were also previously developed by Lee et al. (2018a) and Dai et al. (2017) to generate OOD data or outliers without pre-training. These methods present a GAN-based model with generator and discriminator components, alongside additional density estimation methods, such as a confidence classifier and pre-trained models for OOD prediction. These methods are shown to be not comparable to the OOD exposure-based detector (Du et al., 2023). In contrast, the key adversarial aspect of our framework is that the LGM generates embeddings resembling ID data, while the detector and classifier amplify the energy score gap between these generated pseudo-OOD embeddings and ID samples, implicitly affecting the embeddings' distribution.

### 3.4 ALTERNATING OPTIMISATION

To facilitate effective optimisation of the implicit adversarial objective in Eq. 14, we propose an alternating training schema for GOLD, shown in Algorithm 1. The intuition is to repetitively generate samples close to the ID embeddings of GNN, and then diverge the energy score distribution of the ID and the synthesised pseudo-OOD data. This process ensures the pseudo-OOD data do not diverge too far from ID, enabling effective pseudo-OOD exposure.

Thus, GOLD consists of two alternating optimisation steps: **Step ❶**: **LGM mimicking embeddings of evolving GNN**: A latent generative model is trained to mimic the ID embeddings $\mathbf{H}$, extracted from the $(L-1)$-th layer of an evolving GNN encoder trained in Step 2, to generate pseudo-OOD embeddings. This ensures the pseudo-OOD, $\mathbf{H}_{\text{p-OOD}}$, is suffi-

---

**Algorithm 1** Adversarial Optimisation of GOLD

**Input:** ID graph $\mathcal{G} = (\mathbf{A}, \mathbf{X})$, randomly initialised GNN, MLP detector, and latent generator $D$, epoch numbers $M_1$ and $M_2$, loss coefficients $\lambda, \mu, \gamma$.
**Output:** Optimised GNN, MLP detector, and generative model $D$
1: **while** $train$ **do**
2:    Obtain $\mathbf{H}$ for ID data with GNN from Eq. 7
3:    **for** $epoch = 1, \ldots, M_1$ **do**          **// Step ❶**
4:       Train $D$ with $\mathcal{L}_{\text{Gen}}$ and $\mathbf{H}$ from Eq. 8
5:    **end for**
6:    Sample noise $\mathbf{H}_N$ from Normal distribution
7:    Generate pseudo-OOD $\mathbf{H}_{\text{p-OOD}}$ with $D$ and $\mathbf{H}_T$ from Eq. 9
8:    **for** $epoch = 1, \ldots, M_2$ **do**          **// Step ❷**
9:       Train GNN with $\mathcal{L}_{\text{CLS}}$ and $\mathbf{H}$ Eq. 6
10:       Train GNN and MLP with $\mathcal{L}_{\text{Div}}$, $\mathbf{H}$ and $\mathbf{H}_{\text{p-OOD}}$ from Eq. 13
11:    **end for**
12: **end while**

---

ciently close to the ID embeddings and avoids far away and meaningless generation before subsequent divergence (Line 2-7 in Algorithm 1); **Step ❷**: **Detector and GNN diverging energy of evolving generator**: The GNN encoder and OOD detector are trained to diverge the energy between the ID embeddings and the pseudo-OOD embeddings generated by the evolving LGM trained in Step 1 (Line 8-11 in Algorithm 1). Figure 3 illustrates the adversarial essence of this training paradigm.

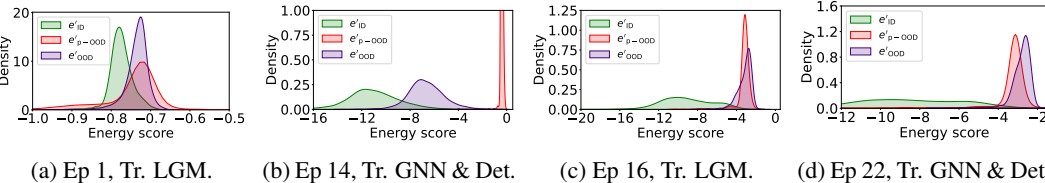

(a) Ep 1, Tr. LGM.     (b) Ep 14, Tr. GNN & Det.     (c) Ep 16, Tr. LGM.     (d) Ep 22, Tr. GNN & Det.

Figure 3: Transformed energy $\mathbf{e}'$ distribution during adversarial training (Tr.) on the `Twitch` dataset for in-distribution (ID), pseudo (p-)OOD, and real OOD across epochs. **(a)** shows that after LGM trains to mimic ID data, energy scores are overlapped for ID, p-OOD, and OOD in the initial stages. **(b)** indicates that after training GNN and the detector to separate the energy of ID and the p-OOD, the real OOD energy cannot be effectively separated from ID. This is a similar situation to the OOD exposure for GNNSafe as in Figure 4c. **(c)** shows that under adversarial learning, the LGM will be updated to generate p-OOD closer to the updated ID data, preventing it from being too far away from ID data with ineffective OOD learning. **(d)** displays the final energy distribution after convergence, with real OOD and ID being well separated, while p-OOD and OOD being well aligned.

Table 1: Model performance comparison: out-of-distribution detection results are measured by **AU-ROC** (↑) / **AUPR** (↑) / **FPR95** (↓) (%) and in-distribution classification results are measured by accuracy (**ID ACC**) (↑). The average performance of the OOD test sets is reported, with variance reflecting performance differences across distinct test sets. Detailed results for individual subsets are reported in Appendix A.9. OOD detection performance was prioritised, with the detection results of our Non-OOD exposed GOLD against Non- (Real-) OOD Exposure methods highlighted by best and runner-up (**best** and runner-up), respectively. Dashed line indicates unavailability.

| | Metrics | Non-OOD Exposure | | | | | | | | Real OOD Exposure | | | | GOLD (Non-OOD) | |
| | | MSP | ODIN | Maha | Energy | GKDE | GPN | GNNSAFE | NODESAFE | OE | Energy FT | GNNSAFE++ | NODESAFE++ | w/ VAE | w/ LDM |
|---|---|---|---|---|---|---|---|---|---|---|---|---|---|---|---|
| **Twitch** | AUROC | 33.59 | 58.16 | 55.68 | 51.24 | 46.48 | 51.73 | 66.82 | 89.99 | 55.72 | 84.50 | 95.36 | 98.50 | 99.26 | **99.46 ± 0.09** |
| | AUPR | 49.14 | 72.12 | 66.42 | 60.81 | 62.11 | 66.36 | 70.97 | 93.33 | 70.18 | 88.04 | 97.12 | 99.18 | 98.54 | **99.62 ± 0.06** |
| | FPR95 | 97.45 | 93.96 | 90.13 | 91.61 | 95.62 | 95.51 | 76.24 | 47.00 | 95.07 | 61.29 | 33.57 | 3.43 | 3.03 | **1.78 ± 0.43** |
| | ID ACC | 68.72 | 70.79 | 70.51 | 70.40 | 67.44 | 68.09 | 70.40 | 71.79 | 70.73 | 70.52 | 70.18 | 71.85 | 68.50 | 68.49 ± 0.13 |
| **Cora** | AUROC | 82.55 | 49.87 | 54.74 | 83.09 | 69.54 | 84.56 | 91.25 | 94.39 | 79.76 | 85.13 | 92.98 | 95.36 | 89.96 | **95.84 ± 0.69** |
| | AUPR | 65.82 | 26.08 | 34.43 | 66.21 | 46.09 | 68.02 | 82.62 | 86.01 | 64.93 | 67.89 | 84.93 | 88.08 | 93.19 | **91.17 ± 2.59** |
| | FPR95 | 62.39 | 100.00 | 96.30 | 65.21 | 80.51 | 58.30 | 47.38 | 26.04 | 75.22 | 51.03 | 38.44 | 20.20 | 28.66 | **17.83 ± 3.78** |
| | ID ACC | 79.91 | 79.61 | 79.57 | 80.34 | 79.86 | 81.65 | 80.37 | 81.92 | 77.69 | 80.44 | 81.45 | 81.65 | 76.79 | 81.66 ± 7.94 |
| **Amazon** | AUROC | 96.52 | 80.12 | 73.81 | 96.73 | 66.98 | 92.60 | 98.49 | - | 97.79 | 98.04 | **98.99** | - | 98.68 | 98.81 ± 1.40 |
| | AUPR | 95.01 | 77.18 | 72.35 | 95.16 | 71.18 | 90.50 | 98.62 | - | 97.26 | 96.96 | 98.88 | - | 98.89 | **98.92 ± 1.31** |
| | FPR95 | 13.83 | 85.22 | 83.44 | 13.15 | 98.47 | 32.64 | 2.30 | - | 7.52 | 5.98 | 2.10 | - | 5.11 | **2.07 ± 3.46** |
| | ID ACC | 93.83 | 93.88 | 93.80 | 93.85 | 87.71 | 89.54 | 93.70 | - | 93.54 | 93.38 | 93.48 | - | 89.91 | 92.99 ± 1.90 |
| **Coauthor** | AUROC | 95.74 | 51.71 | 82.02 | 96.64 | 69.24 | 69.89 | 98.82 | - | 97.65 | 98.17 | **99.28** | - | 98.78 | 99.00 ± 1.19 |
| | AUPR | 96.43 | 56.37 | 87.05 | 97.09 | 80.17 | 72.77 | 99.44 | - | 98.04 | 98.51 | **99.73** | - | 96.40 | 99.56 ± 0.43 |
| | FPR95 | 21.37 | 99.97 | 48.09 | 15.49 | 97.04 | 69.60 | 4.28 | - | 10.61 | 7.76 | 3.18 | - | 4.66 | **3.16 ± 5.46** |
| | ID ACC | 93.37 | 93.29 | 93.29 | 93.57 | 87.74 | 89.39 | 93.56 | - | 93.41 | 93.44 | 93.68 | - | 92.22 | 92.69 ± 1.87 |
| **Arxiv** | AUROC | 63.91 | 55.07 | 56.92 | 64.20 | 58.32 | OOM | 71.06 | 72.44 | 69.80 | 71.56 | 74.77 | 75.49 | 71.52 | 73.90 ± 0.11 |
| | AUPR | 75.85 | 68.85 | 69.63 | 75.78 | 72.62 | OOM | 80.44 | 81.51 | 80.15 | 80.47 | 83.21 | 83.71 | 80.25 | 82.52 ± 0.12 |
| | FPR95 | 90.59 | 100.0 | 94.24 | 90.80 | 93.84 | OOM | 87.01 | 84.27 | 85.16 | 80.59 | 77.43 | 75.24 | 81.95 | 80.57 ± 0.32 |
| | ID ACC | 53.78 | 51.39 | 51.59 | 53.36 | 50.76 | OOM | 53.39 | 51.20 | 52.39 | 53.26 | 53.50 | 52.93 | 49.70 | 50.59 ± 0.53 |

## 4 EXPERIMENTS

**Datasets.** Following Wu et al. (2023b), five benchmark datasets are used for OOD detection evaluation, including four single-graph datasets: (1) `Cora`, (2) `Amazon-Photo`, (3) `Coauthor-CS`, with synthetic OOD data created via: structure manipulation, feature interpolation, and label leave-out; and (4) `ogbn-Arxiv`, OOD by year, and (5) one multi-graph scenario: `TwitchGamers-Explicit`, OOD by different graphs. Detailed splits are provided in Appendix A.6.

**Baselines.** We compared GOLD with 12 baseline models, classed into three categories. (1) General non-OOD exposed methods: MSP (Hendrycks & Gimpel, 2017b), ODIN (Liang et al., 2018), Mahalanobis (short for Maha) (Lee et al., 2018b), and Energy (Liu et al., 2020), with GNN used as backbone. (2) Graph-specific non-OOD exposed detection methods: GKDE (Zhao et al., 2020), GPN (Stadler et al., 2021), GNNSAFE (Wu et al., 2023b), and NODESAFE (Yang et al., 2024). (3) Real OOD exposed methods: adopts techniques from computer vision, such as OE (Hendrycks et al., 2019) and Energy FT (Liu et al., 2020), along with the state-of-the-arts GNNSafe++ (Wu et al., 2023b) and NODESAFE++ (Yang et al., 2024) for graph data. Note that OOD synthesis methods from computer vision (Du et al., 2022; 2023; Lee et al., 2018a; Tao et al., 2023) are not compared due to the non-trivial application from image to graph.

**Metrics.** The following common practice metrics are used for evaluation: AUROC, AUPR, and FPR95 for OOD detection and Accuracy for ID classification. Metric details are in Appendix A.7.

**Implementations.** For a fair comparison, GCN is used as the backbone across all methods, with a layer depth of 2 and a hidden size of 64. The propagation iteration $k$ in Eq. 5 is set to 2, and the controlling parameter $\alpha$ of 0.5 is used. For LDM, the timestep $T$ is configured within $\{600, 800, 1000\}$, $\beta_1 = 10^{-4}$, and $\beta_T = 0.02$. The denoising network $D$ and the MLP detector model are implemented with varying layer and hidden dimension sizes within $\{2, 3\}$ and $\{128, 256, 512\}$ respectively, subject to the dataset. Additional hyperparameter analysis and parameter details are provided in Appendix A.11. We use the Adam optimizer for optimisation (Kingma & Ba, 2015).

### 4.1 OVERALL PERFORMANCE

**Our Non-OOD exposed GOLD can outperform Non-OOD exposure methods and is competitive with Real OOD exposed methods.** As shown in Table 1, GOLD with LDM consistently surpasses the state-of-the-art non-OOD exposure methods NODESAFE and GNNSAFE by a large margin across all datasets, as indicated by the teal colouring. When using VAE as LGM, the OOD detection performance is very close while being more lightweight due to the model design. GOLD

with VAE can achieve state-of-the-art performance especially when the datasets are challenging for general methods, like `Twitch` and `Arxiv`. Additionally, considering LDM as the generative model, GOLD can largely outperform GNNSᴀꜰᴇ++ and achieves better performance than the SOTA NODESᴀꜰᴇ++ in OOD detection for `Twitch` and `CORA`, as highlighted in bold. While for `Amazon`, `Coauthor`, and `Arxiv` dataset, GOLD can achieve a comparable performance with GNNSᴀꜰᴇ++ while not significantly surpassing them. The reason can be two-fold. For `Amazon` and `Coauthor`, the classifier and the OOD detector are already in high performance, which leads to the fact that the energy from the classifier and the information given by the real OOD data have already been well utilised. The pseudo-OOD generation in GOLD cannot provide much more useful supervision signals for the detector. Nonetheless, GOLD still largely outperforms the non-OOD exposure. While for the `Arxiv` dataset, the OOD situation is defined by time, which leads to a huge boost of OOD information when exposing a real OOD dataset. In contrast, for GOLD, the pseudo-OOD generation is largely limited by the ID accuracy of the classifier at 50%. A more detailed table with individual OOD test set performance and variance can be found in Appendix A.9.

Since GOLD uses GNNSᴀꜰᴇ as the backbone, the following detailed experiments are mainly conducted based on the comparison with the base GNNSᴀꜰᴇ/++ approach. GOLD with LDM is used as the default model without specific notation.

## 4.2 VISUALISATION OF ENERGY SCORE GAP

This experiment presents the energy distribution of GOLD and GNNSᴀꜰᴇ. Figures 4a and 4d display **a distinct separation in the energy scores of ID and p-OOD, as well as ID and OOD, produced by the detector**, exemplifying the effectiveness of GOLD in distinguishing and amplifying the energy margin between ID and (p-)OOD data. Furthermore, Figures 4b and 4e illustrate the energy score distributions of the test ID data, synthetic OOD data, and the test OOD data. These figures reveal an optimal and almost disjoint between ID and OOD data, where the thresholds $t_{ID}$ and $t_{OOD}$ indicate a clear energy boundary, thereby indicating the efficacy of GOLD in simulating pseudo-OOD data to facilitate effective OOD detection. Compared with the energy distribution of test ID data, test OOD data and exposed OOD data from GNNSᴀꜰᴇ in Figures 4c and 4f, GOLD can further separate the energy scores between the test ID and OOD data with the pseudo-OOD data.

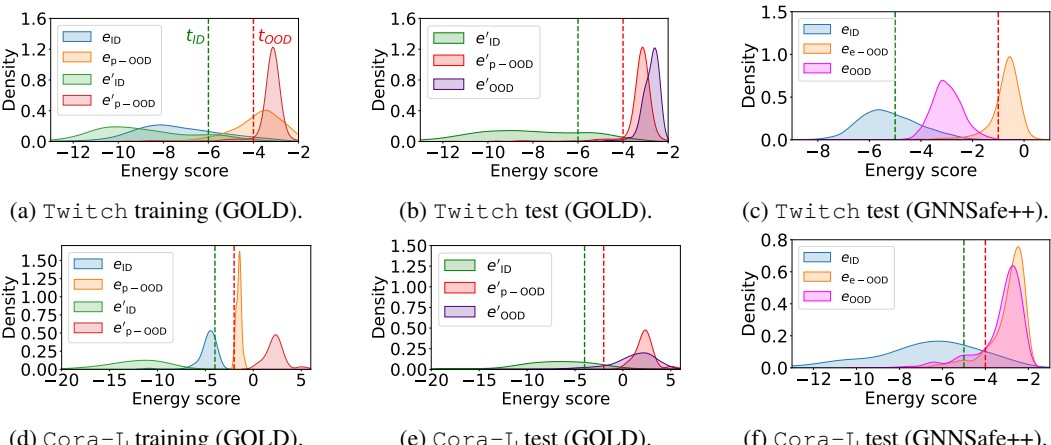

Figure 4: Energy score distributions for `Twitch` and `Cora-L` with GOLD and GNNSᴀꜰᴇ++. The vertical green (red) dashed lines represent the thresholds $t_{ID}$ ($t_{OOD}$) from Eq. 4. **e** denotes original energy scores from the GNN, while **e**′ are transformed scores from the detector, with subscripts for ID, OOD, p-OOD (pseudo), and e-OOD (exposed) data. **(a) & (d)** show that transformed energy **e**′ (green and red) can be further diverged from the original energy **e** (blue and orange). **(b) & (e)** indicate that GOLD can align the transformed energy **e**′ for pseudo OOD (red) and real OOD (purple) in testing. At the same time, the transformed energy **e**′ of ID (green) can be separated. **(c) & (f)** demonstrate that energy separation of test ID (blue) and OOD (pink) in GNNSᴀꜰᴇ++ is not effective, such that although the exposed OOD (orange) can diverge far away from the ID (blue), the real OOD (pink) is still closer to the ID (blue).

## 4.3 ABLATION STUDY

In the ablation study, two variants are studied: (1) pre-training the LDM without the adversarial pipeline (w/o Adv.), and (2) removing the MLP detector, using GNN energy scores instead (w/o Det.). In Table 2, **both the adversarial training paradigm and the new detector significantly contribute to the GOLD**. The results reveal that without adver-

Table 2: Ablation study.

|  | Metrics | GNNSAFE | GNNSAFE++ | w/o Adv. | w/o Det. | GOLD |
|---|---|---|---|---|---|---|
| Twitch | AUROC | 66.82 | 95.36 | 84.59 | 77.70 | **99.46** |
|  | AUPR | 70.97 | 97.12 | 88.69 | 83.91 | **99.62** |
|  | FPR95 | 76.24 | 33.57 | 59.71 | 79.84 | **1.78** |
|  | ID ACC | 70.40 | 70.18 | 70.97 | 70.97 | 68.49 |
| Cora | AUROC | 91.25 | 92.98 | 89.64 | 93.43 | **95.84** |
|  | AUPR | 82.62 | 84.93 | 80.22 | 86.78 | **91.17** |
|  | FPR95 | 47.38 | 38.44 | 46.33 | 34.01 | **17.83** |
|  | ID ACC | 80.37 | 81.45 | 77.60 | 80.70 | 81.66 |
| Arxiv | AUROC | 71.06 | **74.77** | 69.76 | 69.91 | 73.90 |
|  | AUPR | 80.44 | **83.21** | 78.93 | 79.05 | 82.52 |
|  | FPR95 | 87.01 | **77.43** | 88.16 | 89.67 | 80.57 |
|  | ID ACC | 53.39 | 53.50 | 49.89 | 49.66 | 50.59 |

sarial learning, the OOD detection performance has a significant drop for all situations. This underscores the efficacy of the adversarial framework in pseudo-OOD exposure. Moreover, we observe a more unstable performance when removing the detector. In this scenario, the model can still surpass other baselines on datasets like Cora, but it shows a significant drop on others. This juxtaposition exemplifies the necessity of the detector model in GOLD. Nonetheless, the results illustrate the importance of integrating all components to enhance the model's OOD detection capabilities. We provide additional explanation and visualisation of the ablation study in Appendix A.10.

## 4.4 ADVERSARIAL TRAINING ANALYSIS

To further assess the proposed adversarial training framework, three variants of (pseudo-) OOD exposure are studied: (1) using an ID-pretrained LDM to generate once to train GOLD (Gen. Once), which is the same as w/o Adv. in Section 4.3; (2) using an ID-pretrained LDM to generate multiple rounds of pseudo-OOD along the GOLD training loops (Gen.

Table 3: Adversarial training analysis.

|  | Metrics | GNNSAFE++ | Gen. Once | Gen. Multi | Real OOD | GOLD |
|---|---|---|---|---|---|---|
| Twitch | AUROC | 95.36 | 84.59 | 84.33 | 97.58 | **99.46** |
|  | AUPR | 97.12 | 88.69 | 88.38 | 98.50 | **99.62** |
|  | FPR95 | 33.57 | 59.71 | 57.00 | 14.39 | **1.78** |
|  | ID ACC | 70.18 | 70.97 | 71.12 | 70.45 | 68.49 |
| Cora | AUROC | 92.98 | 89.64 | 92.83 | 95.59 | **95.84** |
|  | AUPR | 84.93 | 80.22 | 85.09 | 90.05 | **91.17** |
|  | FPR95 | 38.44 | 46.33 | 30.11 | 21.54 | **17.83** |
|  | ID ACC | 81.45 | 77.60 | 80.56 | 78.42 | 81.66 |
| Arxiv | AUROC | 74.77 | 69.76 | 72.15 | **78.90** | 73.90 |
|  | AUPR | 83.21 | 78.93 | 80.57 | **85.46** | 82.52 |
|  | FPR95 | 77.43 | 88.16 | 82.02 | **68.94** | 80.57 |
|  | ID ACC | 53.50 | 49.89 | 50.77 | 49.99 | 50.59 |

Multi); (3) using real OOD data instead of pseudo-OOD to train GOLD (Real OOD). The results are detailed in Table 3, which highlights that **using an ID-pre-trained generative model would not improve OOD detection performance**. This is because without the adversarial training, the detector will be biased by a set of inaccurate and close-to-ID pseudo-OOD data generated by the pre-trained diffusion model. When incorporating real OOD data to substitute the pseudo-OOD in our framework, the Real OOD variant can achieve consistently better performance than Gen. Once and Gen. Multi. For Arxiv, Real OOD can surpass our default GOLD model with the advantage of OOD exposure in this dataset. Furthermore, a comparison of the results after removing the adversarial process highlights the superiority of the adversarial framework, as all adversarial-based methods outperform their non-adversarial baselines. This robust set of results validates the efficacy of our adversarial training paradigm in enhancing model performance for OOD detection. Despite these modifications, our synthetic-based OOD detection continues to maintain strong performance.

## 4.5 EFFECTIVENESS OF ENERGY REGULARISER

Extending beyond the previous analysis, we observed that the energy regularisers in GOLD are important factors for OOD detection, especially for the divergence regularisation. We provide a comprehensive assessment of the energy regularisers, $\mathcal{L}_{\text{Unc}}$ from Eq. 10, $\mathcal{L}_{\text{EReg}}$ from Eq. 4, and $\mathcal{L}_{\text{DReg}}$ from Eq. 12, across three datasets: Twitch, Cora, and Amazon, reporting the average performance across subsets in Table 4. **The default GOLD that incorporates all regularisers, consistently shows superior performance across all datasets, effectively indicating the contribution of the energy regularisers in OOD detection**. Notably, each dataset exhibits different sensitivities to the absence or presence of specific regularisers. For instance, all datasets are significantly affected by the removal of $\mathcal{L}_{\text{DReg}}$, highlighting its critical role. There is a substantial performance drop for Cora without $\mathcal{L}_{\text{EReg}}$. Additionally, individual regulariser performance is context-dependent, with $\mathcal{L}_{\text{DReg}}$ emerging as particularly impactful, often driving better outcomes when combined with either of the other two regularisers. This is reflected in the best runner-up results, where $\mathcal{L}_{\text{DReg}}$ is combined with another regulariser, underscoring its influence as the most impactful of the three. Nonetheless, this analysis demonstrates the effectiveness of a holistic approach of combining all proposed regularisers, as shown by GOLD's consistently high performance across all metrics and datasets. The extended performance of each subset is provided in Appendix A.9.

Table 4: Energy regulariser analysis.

| $\mathcal{L}_{Unc}$ | $\mathcal{L}_{EReg}$ | $\mathcal{L}_{DReg}$ | Twitch | | | | Cora | | | | Amazon | | | |
|---|---|---|---|---|---|---|---|---|---|---|---|---|---|---|
| | | | AUROC | AUPR | FPR | ID Acc | AUROC | AUPR | FPR | ID Acc | AUROC | AUPR | FPR | ID Acc |
| | | | 86.44 | 80.64 | 79.84 | 68.97 | 61.14 | 57.82 | 89.70 | 76.23 | 64.17 | 72.67 | 46.72 | 92.07 |
| ✓ | | | 10.18 | 40.62 | 97.84 | 70.15 | 70.76 | 65.12 | 94.37 | 81.05 | 67.20 | 71.73 | 68.13 | 93.70 |
| | ✓ | | 78.02 | 83.37 | 78.90 | 70.98 | 66.00 | 63.32 | 54.06 | 80.44 | 48.65 | 61.93 | 77.91 | 93.45 |
| | | ✓ | 69.04 | 76.88 | 44.54 | 70.79 | 84.39 | 74.57 | 68.49 | 76.08 | 97.72 | 96.83 | 8.28 | 92.79 |
| ✓ | ✓ | | 76.88 | 81.49 | 76.14 | 70.99 | 34.63 | 39.27 | 96.84 | 81.25 | 71.15 | 63.84 | 74.85 | 93.30 |
| ✓ | | ✓ | 64.43 | 75.46 | 45.95 | 70.90 | 94.03 | 73.89 | 73.54 | 74.53 | 97.91 | 97.03 | 4.54 | 93.20 |
| | ✓ | ✓ | 89.58 | 93.12 | 43.78 | 69.64 | 93.28 | 87.50 | 31.04 | 79.88 | 98.02 | 98.42 | 3.40 | 92.81 |
| GOLD | | | 99.46 | 99.62 | 1.78 | 68.49 | 95.84 | 91.17 | 17.83 | 81.66 | 98.81 | 98.92 | 2.07 | 92.99 |

## 4.6 COMPUTATIONAL COST

Table 5 shows that **GOLD generally achieves a very close inference time, and a faster (w/ VAE) or comparable (w/LDM) training time relative to the GNNSafe(++) baseline**.

Table 5: Inference and training time (s) of GOLD.

| | GNNSAFE | | GNNSAFE++ | | GOLD w/ VAE | | GOLD w/ LDM | |
|---|---|---|---|---|---|---|---|---|
| | Inf. | Train. | Inf. | Train. | Inf. | Train. | Inf. | Train. |
| Twitch | 0.08 | 2.41 | 0.09 | 4.74 | 0.09 | 2.78 | 0.10 | 8.96 |
| Cora-F | 0.03 | 4.40 | 0.03 | 5.32 | 0.04 | 3.91 | 0.04 | 5.93 |
| Amazon-F | 0.04 | 13.51 | 0.05 | 18.40 | 0.05 | 12.52 | 0.07 | 39.04 |
| Coauthor-F | 0.35 | 57.80 | 0.36 | 67.83 | 0.35 | 55.65 | 0.37 | 89.74 |
| Arxiv | 0.40 | 85.23 | 0.40 | 132.36 | 0.45 | 80.77 | 0.47 | 244.95 |

This is under the situation that the non-OOD exposed GOLD outperforms the existing non-OOD exposure methods, while matching or surpassing the real-OOD exposed SOTA baselines, all under the same backbone. In addition to the high-performing LDM variant, a lightweight VAE is also experimented, providing an efficient alternative with comparable performance. Thus, we consider this training cost as an acceptable trade-off for improved OOD detection performance, and is discussed in Appendix A.4. However, we highlight that GOLD can achieve a similar inference time as the baselines, regardless of the LGM, as shown in Table 5. This reveals a competitive application of GOLD while having a strong performance. We provide detailed results in Appendix A.12.

## 5 RELATED WORK

Our work intersects with three major research areas: **1) Non-OOD-Exposure OOD Detection** that purely relies on ID data for detecting OOD instances, this involves score-based methods, feature learning, and techniques specific for graph-structured data (Lee et al., 2018a; Hendrycks & Gimpel, 2017a;b; Koo et al., 2024; Liu et al., 2020; Ding & Shi, 2023; Ma et al., 2023; Zhao et al., 2020; Liu et al., 2023b; Li et al., 2022b; Wu et al., 2023b; Yang et al., 2024); **2) OOD Exposure-Based OOD Detection**, a prominent line of work that adopts auxiliary OOD data to assist training, often achieving higher performance than non-OOD-exposure based methods (Hendrycks et al., 2019; Liu et al., 2020; Park et al., 2023; Zhu et al., 2023; Zheng et al., 2023; Du et al., 2024; Wu et al., 2023b; Bao et al., 2024); and **3) OOD Generation,** a more recent field that aims to synthesise OOD-like data to assist OOD detection (Vernekar et al., 2019; Serrà et al., 2020; Xiao et al., 2020; Wang et al., 2020; Nalisnick et al., 2019; Schirrmeister et al., 2020; Lee et al., 2018a; Du et al., 2022; Tao et al., 2023). Notably for graph data, GNNSAFE considers the inter-dependence nature of node instances and proposes an energy propagation schema, and explores an OOD-exposed variant GNNSAFE++ (Wu et al., 2023b). NODESAFE/++ builds upon GNNSAFE/++ and proposes additional regularisation terms to reduce and bound the generation of extreme energy scores (Yang et al., 2024). Bao et al. (2024) proposes a generalised Dirichlet energy score for graph OOD detection. A detailed review of related work is provided in Appendix A.3.

## 6 CONCLUSION

In this paper, we propose GOLD, a novel graph OOD detection framework with a latent generative model trained in a novel implicit adversarial paradigm. Unlike methods that rely on pre-trained generative models or real OOD data requiring auxiliary data inputs, GOLD synthesises pseudo-OOD data to inherit OOD characteristics through the implicit adversarial framework, solely based on ID data. An effective OOD detector head is further designed to address the difficulties with multiple classes in the logit space, optimising the energy score for improved detection. Extensive experiments show the efficacy of GOLD, outperforming SOTA non- and OOD-exposed methods. We hope this work inspires future synthetic-based graph OOD detection research for real-world applications.

## 7 ACKNOWLEDGEMENT

This research has been partially supported by Australian Research Council Discovery Projects (DP230101196, DP24010306, DE250100919 and CE200100025).

## 8 REPRODUCIBILITY STATEMENT

To support reproducible research, we summarise our efforts as below:

1. **Baselines & Datasets.** We follow the baseline from (Wu et al., 2023b) and utilise publicly available datasets. The details are described in Section 4 and Appendix A.6.

2. **Model training.** Our implementation of the energy-based OOD detector builds upon the open-sourced work GNNSAFE by Wu et al. (2023b), `https://github.com/qitianwu/GraphOOD-GNNSafe`. Detailed implementation setting is provided in Section 4 and Appendix A.8.

3. **Methodology.** Our GOLD framework is fully documented in Section 3. In addition, we provide a detailed pseudo code in Algorithm 1.

4. **Evaluation Metrics.** We discuss the evaluation metrics used in Section 4 and Appendix A.7.

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

# A  APPENDIX

In the Appendix, we provide additional supplementary material to the main paper. The structure is as follows:

- We provide the proof for Proposition 1. in A.1.

- An extended related work is detailed in A.3.

- Potential Limitations is discussed in A.4.

- Preliminary GNN description is described in A.5.

- We provide the description of datasets in A.6.

- The evaluation metrics and implementation details are provided in A.7 and A.8.

- Additional experiment results, including extended subset performance, ablation study visualisations, empirical evaluations of logits vs. softmax scores, and computational cost are detailed in A.9, A.10 A.2, A.12.

- Descriptions of the latent generative models: 1) Latent diffusion model, and 2) Variational autoencoder are provided in A.11.

## A.1  PROOF FOR PROPOSITION 1.

*Proof.* Let $l_{\theta_{[y]}}$ denote the logits of the MLP detector with parameter $\theta$ for class y, $\phi$ denote the softmax function. Assume the hyper-parameters $\lambda = \gamma = 1$.

Note that:

$$\frac{\partial \log(e_\theta^a + e_\theta^b)}{\partial \theta} = \frac{e_\theta^a \frac{\partial a_\theta}{\partial \theta} + e_\theta^b \frac{\partial b_\theta}{\partial \theta}}{e_\theta^a + e_\theta^b} \tag{15}$$

The gradient of $\min_{\text{MLP}} -(\mathcal{L}_{\text{Unc}} + \mathcal{L}_{\text{DReg}})$ w.r.t $\theta$ is given by:

$$
\begin{aligned}
-\frac{\partial \mathcal{L}_{\text{Unc}}}{\partial \theta} &= -\mathbb{E}_{i \sim P_{\text{ID}}} \frac{\partial \log[\phi(l_\theta(e_i))_{[0]}]}{\partial \theta} - \mathbb{E}_{j \sim P_{\text{p-OOD}}} \frac{\partial \log[\phi(l_\theta(e_j))_{[1]}]}{\partial \theta} \\
&= -\mathbb{E}_{i \sim P_{\text{ID}}} \frac{\partial \log[\frac{e^{l_\theta(e_i)_{[0]}}}{e^{l_\theta(e_i)_{[0]}} + e^{l_\theta(e_i)_{[1]}}}]}{\partial \theta} - \mathbb{E}_{j \sim P_{\text{p-OOD}}} \frac{\partial \log[\frac{e^{l_\theta(e_j)_{[1]}}}{e^{l_\theta(e_j)_{[0]}} + e^{l_\theta(e_j)_{[1]}}}]}{\partial \theta} \\
&= -\mathbb{E}_{i \sim P_{\text{ID}}} \frac{\partial [l_\theta(e_i)_{[0]} - \log(e^{l_\theta(e_i)_{[0]}} + e^{l_\theta(e_i)_{[1]}})]}{\partial \theta} \\
&\quad - \mathbb{E}_{j \sim P_{\text{p-OOD}}} \frac{[\partial l_\theta(e_j)_{[1]} - \log(e^{l_\theta(e_j)_{[0]}} + e^{l_\theta(e_j)_{[1]}})]}{\partial \theta} \\
&= \mathbb{E}_{i \sim P_{\text{ID}}} \left[ -\frac{\partial l_\theta(e_i)_{[0]}}{\partial \theta} + \frac{e^{l_\theta(e_i)_{[0]}} \frac{\partial l_\theta(e_i)_{[0]}}{\partial \theta} + e^{l_\theta(e_i)_{[1]}} \frac{\partial l_\theta(e_i)_{[1]}}{\partial \theta}}{e^{l_\theta(e_i)_{[0]}} + e^{l_\theta(e_i)_{[1]}}} \right] \\
&\quad + \mathbb{E}_{j \sim P_{\text{p-OOD}}} \left[ -\frac{\partial l_\theta(e_j)_{[1]}}{\partial \theta} + \frac{e^{l_\theta(e_j)_{[0]}} \frac{\partial l_\theta(e_j)_{[0]}}{\partial \theta} + e^{l_\theta(e_j)_{[1]}} \frac{\partial l_\theta(e_j)_{[1]}}{\partial \theta}}{e^{l_\theta(e_j)_{[0]}} + e^{l_\theta(e_j)_{[1]}}} \right]
\end{aligned}
\tag{16}
$$

Notice that $\max(0, e_i - e_i')$ and $\max(0, e_j' - e_j)$ are positive and monotonic, the optimised $\theta$ that minimises the functions (arg min) would also minimise $\max(0, e_i - e_i')^2$ and $\max(0, e_j' - e_j)^2$, thus, we consider the gradient of a surrogate function of $\mathcal{L}_{\text{DReg}}$ as $\mathcal{L}_{\text{DReg}_S}$:

$$-\frac{\partial \mathcal{L}_{\text{DReg}_S}}{\partial \theta} = -\frac{\partial}{\partial \theta} \mathbb{E}_{i \sim P_{\text{ID}}} \max\left(0, e_i - e'_i\right) - \frac{\partial}{\partial \theta} \mathbb{E}_{j \sim P_{\text{p-OOD}}} \max\left(0, e'_j - e_j\right)$$

If $e_i - e'_i \leq 0$ or $e'_j - e_j \leq 0$ the gradient is 0, else:

$$= -\frac{\partial}{\partial \theta} \mathbb{E}_{i \sim P_{\text{ID}}} \left[ e_i + \log(e^{l_\theta(e_i)_{[0]}} + e^{l_\theta(e_i)_{[1]}}) \right]$$

$$- \frac{\partial}{\partial \theta} \mathbb{E}_{j \sim P_{\text{p-OOD}}} \left[ -\log(e^{l_\theta(e_j)_{[0]}} + e^{l_\theta(e_j)_{[1]}}) - e_j \right]$$

$$= -\mathbb{E}_{i \sim P_{\text{ID}}} \frac{\partial \log(e^{l_\theta(e_i)_{[0]}} + e^{l_\theta(e_i)_{[1]}})}{\partial \theta} \tag{17}$$

$$- \mathbb{E}_{j \sim P_{\text{p-OOD}}} \frac{\partial - \log(e^{l_\theta(e_j)_{[0]}} + e^{l_\theta(e_j)_{[1]}})}{\partial \theta}$$

$$= -\mathbb{E}_{i \sim P_{\text{ID}}} \frac{e^{l_\theta(e_i)_{[0]}} \frac{\partial l_\theta(e_i)_{[0]}}{\partial \theta} + e^{l_\theta(e_i)_{[1]}} \frac{\partial l_\theta(e_i)_{[1]}}{\partial \theta}}{e^{l_\theta(e_i)_{[0]}} + e^{l_\theta(e_i)_{[1]}}}$$

$$+ \mathbb{E}_{j \sim P_{\text{p-OOD}}} \frac{e^{l_\theta(e_j)_{[0]}} \frac{\partial l_\theta(e_j)_{[0]}}{\partial \theta} + e^{l_\theta(e_j)_{[1]}} \frac{\partial l_\theta(e_j)_{[1]}}{\partial \theta}}{e^{l_\theta(e_j)_{[0]}} + e^{l_\theta(e_j)_{[1]}}}$$

$$-\left(\frac{\partial \mathcal{L}_{\text{Unc}}}{\partial \theta} + \frac{\partial \mathcal{L}_{\text{DReg}_S}}{\partial \theta}\right) = \mathbb{E}_{i \sim P_{\text{ID}}} \left[ -\frac{\partial l_\theta(e_i)_{[0]}}{\partial \theta} + \frac{e^{l_\theta(e_i)_{[0]}} \frac{\partial l_\theta(e_i)_{[0]}}{\partial \theta} + e^{l_\theta(e_i)_{[1]}} \frac{\partial l_\theta(e_i)_{[1]}}{\partial \theta}}{e^{l_\theta(e_i)_{[0]}} + e^{l_\theta(e_i)_{[1]}}} \right]$$

$$+ \mathbb{E}_{j \sim P_{\text{p-OOD}}} \left[ -\frac{\partial l_\theta(e_j)_{[1]}}{\partial \theta} + \frac{e^{l_\theta(e_j)_{[0]}} \frac{\partial l_\theta(e_j)_{[0]}}{\partial \theta} + e^{l_\theta(e_j)_{[1]}} \frac{\partial l_\theta(e_j)_{[1]}}{\partial \theta}}{e^{l_\theta(e_j)_{[0]}} + e^{l_\theta(e_j)_{[1]}}} \right]$$

$$- \mathbb{E}_{i \sim P_{\text{ID}}} \frac{e^{l_\theta(e_i)_{[0]}} \frac{\partial l_\theta(e_i)_{[0]}}{\partial \theta} + e^{l_\theta(e_i)_{[1]}} \frac{\partial l_\theta(e_i)_{[1]}}{\partial \theta}}{e^{l_\theta(e_i)_{[0]}} + e^{l_\theta(e_i)_{[1]}}}$$

$$+ \mathbb{E}_{j \sim P_{\text{p-OOD}}} \frac{e^{l_\theta(e_j)_{[0]}} \frac{\partial l_\theta(e_j)_{[0]}}{\partial \theta} + e^{l_\theta(e_j)_{[1]}} \frac{\partial l_\theta(e_j)_{[1]}}{\partial \theta}}{e^{l_\theta(e_j)_{[0]}} + e^{l_\theta(e_j)_{[1]}}} \tag{18}$$

Define the energy w.r.t. label y as $E(e, y) = -l_\theta(e)_{[y]}$

$$= \mathbb{E}_{i \sim P_{\text{ID}}} \frac{\partial E(e_i, 0)}{\partial \theta} + \mathbb{E}_{j \sim P_{\text{p-OOD}}} \frac{\partial E(e_j, 1)}{\partial \theta}$$

$$- 2\left( \phi(l_\theta(e_j))_{[0]} \frac{\partial E(e_j, 0)}{\partial \theta} + \phi(l_\theta(e_j))_{[1]} \frac{\partial E(e_j, 1)}{\partial \theta} \right)$$

$$= \mathbb{E}_{i \sim P_{\text{ID}}} \frac{\partial E(e_i, 0)}{\partial \theta} + \mathbb{E}_{j \sim P_{\text{p-OOD}}} (1 - 2\phi(l_\theta(e_j))_{[1]}) \frac{\partial E(e_j, 1)}{\partial \theta}$$

$$- 2\phi(l_\theta(e_j))_{[0]} \frac{\partial E(e_j, 0)}{\partial \theta}$$

From the above equation, the training procedure that overall minimises the first order gradient of the negative sum of $\mathcal{L}_{\text{Unc}}$ and the surrogate function of $\mathcal{L}_{\text{DReg}}$ will decrease the energy score $E(e_i, 0; l_\theta)$ for in-distribution data, and increase the energy score $E(e_j, 1; l_\theta)$ and $E(e_j, 0; l_\theta)$ for pseudo-OOD data, given $\phi(l_\theta(e_j))_{[1]} > 0.5$ as the detector continues to improve detection performance. $\qquad\square$

## A.2 LOGITS VS. SOFTMAX DISCREPANCY

In this section, we present an empirical evaluation of the Logits vs. Softmax discrepancy between ID and OOD data from the detector. It is evident from Figure 5, while the softmax confidence scores present high confidence for ID and OOD instances, where the majority of the scores corresponding to the respective class were close to 1 (based on the marginal distribution), the energy scores provide more meaningful information for distinguishing between them. Notably, ID data typically possess higher and positive ID Logits and lower OOD logits than OOD data. Thereby, leading to more distinguishable energy scores than softmax for OOD detection.

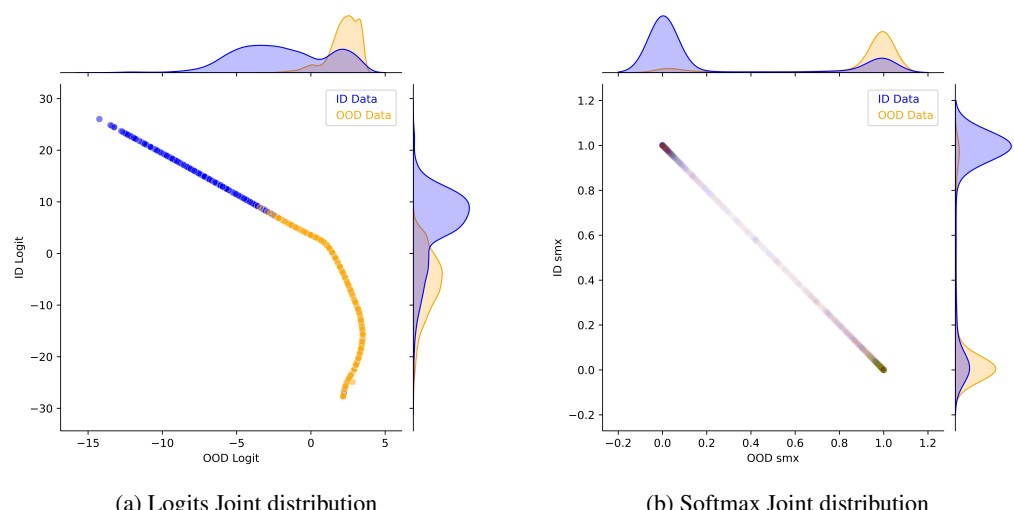

(a) Logits Joint distribution

(b) Softmax Joint distribution

Figure 5: Logits vs. Softmax joint distribution plot for Twitch dataset.

## A.3 EXTENDED RELATED WORK

**Non-OOD-Exposure OOD Detection.** OOD detection is a fundamental task extensively studied in diverse machine learning domains (Lee et al., 2018a; Lafon et al., 2023; Jiang et al., 2023; Huang et al., 2022c; Hendrycks & Gimpel, 2017b; Lee et al., 2018b; Koo et al., 2024; Papadopoulos et al., 2021; Chen et al., 2023). A representative line of work that relies on purely ID data is based on the model's output including using softmax score (Hendrycks & Gimpel, 2017a; Liang et al., 2018), using energy score (Liu et al., 2020; Wang et al., 2021; Yang et al., 2024), and activation pruning-based methods (Djurisic et al., 2023; Sun & Li, 2022; Sun et al., 2021). Other approaches involve confidence enhancement (Hsu et al., 2020; Hein et al., 2019; Vyas et al., 2018), feature learning (Lin et al., 2021; Dong et al., 2022), and adversarial strategies (Bitterwolf et al., 2020; Chen et al., 2021; Choi & Chung, 2020). More recent studies have applied OOD detection to graph-structured data (Ding & Shi, 2023; Ma et al., 2023; Wang et al., 2024; Huang et al., 2022a;b; Trivedi et al., 2024; Yang et al., 2023b; Li et al., 2024; Wang & Li, 2023; Gui et al., 2022; Bazhenov et al., 2022). For node-level detection, GNNSAFE considers the inter-dependence nature of node instances and proposes an energy propagation schema (Wu et al., 2023b). NODESAFE builds upon GNNSAFE and proposes additional regularisation terms to reduce and bound the generation of extreme energy scores (Yang et al., 2024). GKDE proposes a multi-source uncertainty framework to estimate the node-level Dirichlet distributions to assist OOD detection (Zhao et al., 2020). GPN applies Bayesian posterior and density estimation to estimate the uncertainty for each node (Stadler et al., 2021). For graph-level detection, recent methods include modelling distribution shifts through a graph generative process, overseeing from a data-centric perspective, and unsupervised methods (Li et al., 2022b; Guo et al., 2023; Liu et al., 2023b).

**OOD Exposure-Based OOD Detection.** OOD exposure is another prominent line of work that adopts auxiliary OOD data to assist training (Hendrycks et al., 2019; Liu et al., 2020; Park et al., 2023; Zhu et al., 2023; Zheng et al., 2023; Du et al., 2024; Wu et al., 2023b). The aforementioned GNNSAFE model also considers an additional version GNNSAFE++ to adopt OOD exposure and has shown greater performance than the standard model (Wu et al., 2023b). Yang et al. (2024) also presents NODESAFE++ as an extended OOD exposed version. Bao et al. (2024) proposes a generalised Dirichlet energy score for OOD detection. Our proposed GOLD method attempts to take advantage of the effectiveness of OOD exposure by synthesising samples that exhibit OOD characteristics. Thus, avoiding the necessity of real OOD data during training.

**OOD Generation.** Recent studies begin to work on synthesising OOD data (Vernekar et al., 2019; Serrà et al., 2020; Xiao et al., 2020; Wang et al., 2020; Nalisnick et al., 2019; Schirrmeister et al., 2020; Lee et al., 2018a; Du et al., 2022; Tao et al., 2023). A GAN-based approach is proposed

to generate OOD data by jointly training a confidence classifier (Lee et al., 2018a). VOS generates synthetic outliers from low-probability regions of multivariate Gaussian distributions (Du et al., 2022). Recently,pre-trained diffusion models have been widely employed for OOD generation including DFDD (Wu et al., 2023a), Dream-OOD (Du et al., 2023). Several initial graph-level OOD studies have been initiated, predominantly for molecule (Lee et al., 2023a; Shen et al., 2024). A score-based OOD molecule generation model is proposed by MOOD (Lee et al., 2023a), which employs an OOD-controlled reverse-time diffusion. A recent work PGR-MOOD (Shen et al., 2024) proposes to rely on a pre-trained molecule diffusion for generation. These methods typically rely on pre-trained models that are trained with additional data. In contrast, GOLD does not rely on pre-trained generative models to synthesise pseudo-OOD data.

### A.4 POTENTIAL LIMITATIONS

In our concluding remarks, we highlight that our methodology leverages a generative model (specifically, diffusion model) to generate effective pseudo-OOD instances for OOD detection. To curb computational expenses, we employ a latent diffusion model, which reduces the computational demands of direct input space manipulation. Despite this, training-time efficiency may still be impacted. Nonetheless, during the inference phase, our model does not necessitate the generation of extra data, thus mitigating the impact of high latency. Moreover, we have experimented with a lightweight VAE as the latent generative model, which can achieve a competitive computational time as the standard SOTA baselines. Additionally, our approach currently targets node-level prediction tasks; however, we envisage its applicability to graph-level OOD detection, which we leave for future research. Following the propositions of Liu et al. (2020) and Wu et al. (2023b), our framework incorporates an energy-bounded regulariser that ideally ensures ID scores are lower than those of OOD samples, as illustrated in our visualisations in Section 4.2. Extended experiments detailed in Table 19 reveal that using only the energy regulariser results in AUROC scores near the single-digit range. This outcome highlights the regulariser's limitations and challenges the assumption that OOD energy scores consistently exceed ID scores, thereby undermining the effectiveness of the OOD metric in true detection performance. Nevertheless, our framework introduces an additional regulariser, which effectively addresses these discrepancies, as showcased by our consistently positive results.

### A.5 GNN

GNNs, by their very nature, excel in modelling the complex relationship of node-dependence in graphs. Central to their success is the message-passing mechanism, which iteratively aggregates neighbouring information towards the centre node to capture both local and global knowledge. Denote the learnt representation of node $i$ at the $l$-th layer as $\mathbf{h}_i^{(l)}$, a typical Graph Convolutional Network (GCN) executes recursive layer propagation via:

$$\mathbf{H}^{(l)} = \sigma(\mathbf{D}^{-1/2}\tilde{\mathbf{A}}\mathbf{D}^{-1/2}\mathbf{H}^{(l-1)}\mathbf{W}^{(l)}), \mathbf{H}^{l-1} = [\mathbf{h}_i^{l-1}], \mathbf{H}^{(0)} = \mathbf{X} \tag{19}$$

with $\tilde{\mathbf{A}} = \mathbf{A} + \mathbf{I}$, where $\mathbf{I}$ is the identity matrix, $\mathbf{D}$ is the diagonal degree matrix of $\tilde{\mathbf{A}}$, $\sigma$ is a non-linear activation function (i.e., ReLU), and $\mathbf{W}^{(l)}$ is the corresponding weight matrix at layer $l$ (Kipf & Welling, 2017).

### A.6 DESCRIPTION OF DATASETS

The datasets utilised in this study are publicly available benchmark datasets for graph learning. We follow the same data collection and processing protocol in Wu et al. (2023b) and utilised the data loader for the `ogbn-Arxiv` dataset provided by the OGB package[2], and others from the Pytorch Geometric Package[3]. For all datasets, we follow the provided splits and generation process in Wu et al. (2023b). We provide a brief description of the datasets below:

The `TwitchGamers - Explicit` dataset consists of multiple subgraphs, each representing a social network from a different region (Rozemberczki & Sarkar, 2021). The nodes within these subgraphs indicate Twitch gamers, while the edges depict the follower relationships between two

---

[2]https://github.com/snap-stanford/ogb?tab=readme-ov-file
[3]https://pytorch-geometric.readthedocs.io/en/latest/modules/datasets.html

users. Node features include embeddings based on the games played by Twitch users, and for this study, we focus on the label that indicates whether a user broadcasts mature content (i.e., Explicit). We utilise subgraph DE as ID data, and subgraphs ES, FR, RU as testing data. Dataset details are provided in Table 6.

| Twitch | Splits | # Nodes | # Edges | Feature Dimension | # Classes |
|---|---|---|---|---|---|
| Twitch-DE | ID | 9498 | 315774 | 128 | 2 |
| Twitch-ES | OOD | 4648 | 123412 | 128 | 2 |
| Twitch-FR | OOD | 6551 | 231883 | 128 | 2 |
| Twitch-RU | OOD | 4385 | 78993 | 128 | 2 |

Table 6: `Twitch` dataset overview

The `Cora` dataset is a citation network where each node represents a published paper, and each edge reflects a citation relationship between papers (Sen et al., 2008). The dataset consists of seven labels. Since `Cora` does not contain an explicit domain attribute to partition into OOD subgraphs, we follow the provided protocol in Wu et al. (2023b), and synthetically create the OOD data as mentioned in Section 4. Dataset details are provided in Table 7.

| Cora | Splits | # Nodes | # Edges | Feature Dimension | # Classes |
|---|---|---|---|---|---|
| Cora-S | ID | 2708 | 10556 | 1433 | 7 |
| Cora-S | OOD | 2708 | 6696 | 1433 | 7 |
| Cora-F | ID | 2708 | 10556 | 1433 | 7 |
| Cora-F | OOD | 2708 | 10556 | 1433 | 7 |
| Cora-L | ID | 904 | 10556 | 1433 | 3 |
| Cora-L | OOD | 986 | 10556 | 1433 | 3 |

Table 7: `Cora` dataset overview

The `Amazon-Photo` dataset forms an item co-purchasing network on Amazon, where each node represents a product and each edge signifies that the linked products are frequently bought together (McAuley et al., 2015). Node labels categorise the products. Similar to the `Cora` dataset, we employ three synthetic methods to create the OOD data due to the lack of a clear domain for partition. Dataset details are provided in Table 8.

| Amazon-Photo | Splits | # Nodes | # Edges | Feature Dimension | # Classes |
|---|---|---|---|---|---|
| Amazon-S | ID | 7650 | 238162 | 745 | 8 |
| Amazon-S | OOD | 7650 | 149168 | 745 | 8 |
| Amazon-F | ID | 7650 | 238162 | 745 | 8 |
| Amazon-F | OOD | 7650 | 238162 | 745 | 8 |
| Amazon-L | ID | 3095 | 238162 | 745 | 3 |
| Amazon-L | OOD | 3673 | 238162 | 745 | 4 |

Table 8: `Amazon-Photo` dataset overview

The `Coauthor-CS` dataset describes a network of computer science coauthors. Such that each node represents an author, and edges connect any two authors who have collaborated on a paper. The dataset aims to classify authors into their respective fields of study based on the keywords from their publications, which are also used as node features. Due to the lack of a clear domain to split the data, OOD graphs were constructed following the same protocol aforementioned. Dataset details are provided in Table 9.

| Coauthor-CS | Splits | # Nodes | # Edges | Feature Dimension | # Classes |
|---|---|---|---|---|---|
| Coauthor-S | ID | 18333 | 163788 | 6805 | 15 |
| Coauthor-S | OOD | 18333 | 92802 | 6805 | 15 |
| Coauthor-F | ID | 18333 | 163788 | 6805 | 15 |
| Coauthor-F | OOD | 18333 | 163788 | 6805 | 15 |
| Coauthor-L | ID | 13290 | 163788 | 6805 | 10 |
| Coauthor-L | OOD | 3649 | 163788 | 6805 | 4 |

Table 9: `Coauthor-CS` dataset overview

The `ogbn-Arxiv` dataset curated an extensive dataset from 1960 to 2020, where each node represents a paper, labelled by its subject area for classification (Hu et al., 2020). Edges reflect the citation relationships among papers, and each node is associated with a 128-dimensional vector derived from word embeddings of its title and abstract. Following Wu et al. (2023b), we utilise time information to partition the graph, where data before 2015 are used as ID data and papers published after 2017 are used as OOD data. Dataset details are provided in Table 10.

| ogbn-Arxiv | Splits | # Nodes | # Edges | Feature Dimension | # Classes |
|---|---|---|---|---|---|
| Arxiv-2015 | ID | 53160 | 152226 | 128 | 40 |
| Arxiv-2018 | OOD | 29799 | 622466 | 128 | 40 |
| Arxiv-2019 | OOD | 39711 | 1061197 | 128 | 40 |
| Arxiv-2020 | OOD | 8892 | 1166243 | 128 | 40 |

Table 10: `ogbn-Arxiv` dataset overview

## A.7 EVALUATION METRICS

In this section, we provide a detailed description of the metrics used for evaluation. Following common practice in OOD detection (Wu et al., 2023b; Liu et al., 2020; 2023b), we employed three key metrics to measure the performance of detecting OOD instances: (1) the Area Under the Receiver Operating Characteristic curve (AUROC); (2) the Area Under the Precision-Recall curve (AUPR); and (3) the false positive rate (FPR95) of OOD examples when the true positive rate of ID examples is 95%. AUROC measures the trade-off between the true positive rate (TPR) and the false positive rate (FPR) at different threshold levels, providing insights into the model's ability to accurately distinguish between ID and OOD instances. However, in highly imbalanced datasets with only a few OOD instances, AUROC might be overly optimistic. AUPR, on the other hand, offers a more realistic performance measure by accounting for both precision and recall. FPR95 provides further insight into the model's performance under high-sensitivity conditions, indicating the probability of misclassifying in-distribution samples as OOD when the TPR is 95%.

## A.8 IMPLEMENTATION DETAILS

We utilised the publicly available benchmark (i.e., datasets and baselines) provided by Wu et al. (2023b), and fully respect their CC-BY 4.0 license. The experiments were conducted using Python 3.8.0 and PyTorch 2.2.2 with Cuda 12.1, using Tesla V100 GPUs with 32GB memory for experiments. The datasets were obtained from Pytorch Geometric 2.0.3 and OGB 1.3.3 under the MIT license. Extending beyond the thresholds provided in Wu et al. (2023b), we tuned the margins $t_{\text{ID}}$ and $t_{\text{OOD}}$ with various ranges for different dataset (i.e., for `Twitch` $t_{\text{ID}} \in \{-5, -4, -3\}, t_{\text{OOD}} \in \{1, 2, 3\}$). The detector loss weights $\lambda, \mu, \gamma$ are tuned in the range of $\{0, 0.3, 0.5, 0.7, 1, 1.5\}$, depending on the dataset. Hyperparameter sensitivity analysis for the detector and classifier loss objective can be found in Figure 6. The LGM training step $M_1$ is configured in the range of $\{100, 200, 600, 800\}$, and the classifier and detector update $M_2$ is tuned from $\{5 - 20\}$ subject to the dataset, with early stopping applied to ensure the ID accuracy does not reduce significantly. Regarding baseline models, we utilised the provided benchmark in Wu et al. (2023b), which includes modified versions of the different baseline models. This involves adapting to the same encoder GCN backbone (i.e., a hidden size of 64 and layer number of 2) for MSP, ODIN, Mahalanobis, Energy

and Energy FT. We also considered latest SOTA OOD Detection method by Yang et al. (2024) using their reported results.

## A.9 ADDITIONAL EXPERIMENT RESULTS

In this section, we provide additional experimental results to supplement the results provided in the maintext. Specifically, we present detailed OOD detection performance of the subsets for each OOD dataset (i.e., subgraphs for `Twitch`, three types of OOD data for `Cora`, `Amazon`, and `Coauthor`, and different years for `Arxiv`) in Tables 11 to 15, complementing Table 1 in the main text. Furthermore, in Tables 16 and 17, we report an extended version of the ablation study and adversarial training effectiveness, covering the subsets of `Twitch`, `Cora`, and `Arxiv`, supplementing Table 2 and 3 in the maintext. Lastly, we provide the full tables for the energy regulariser analysis in Table 4 for `Twitch`, `Cora`, and `Amazon` in Tables 18, 19, and 20, respectively.

Table 11: Model performance on OOD sub-graphs ES, FR and RU of `Twitch` dataset.

| Dataset | Metrics | Non-OOD Exposure | | | | | | | Real OOD Exposure | | | Ours |
| | | MSP | ODIN | Mahalanobis | Energy | GKDE | GPN | GNNSAFE | OE | Energy FT | GNNSAFE++ | GOLD |
|---|---|---|---|---|---|---|---|---|---|---|---|---|
| Twitch-ES | AUROC | 37.72 | 83.83 | 45.66 | 38.80 | 48.70 | 53.00 | 49.07 | 55.97 | 80.73 | 94.54 | 99.72 ± 0.03 |
| | AUPR | 53.08 | 80.43 | 58.82 | 54.26 | 61.05 | 64.24 | 57.62 | 69.49 | 87.56 | 97.17 | 99.82 ± 0.02 |
| | FPR95 | 98.09 | 33.28 | 95.48 | 95.70 | 95.37 | 95.05 | 93.98 | 94.94 | 76.76 | 44.06 | 0.44 ± 0.13 |
| | ID ACC | 68.72 | 70.79 | 70.51 | 70.40 | 67.44 | 68.09 | 70.40 | 70.73 | 70.52 | 70.18 | 68.49 ± 0.13 |
| Twitch-FR | AUROC | 21.82 | 59.82 | 40.40 | 57.21 | 49.19 | 51.25 | 63.49 | 45.66 | 79.66 | 93.45 | 99.08 ± 0.19 |
| | AUPR | 38.27 | 64.63 | 46.69 | 61.48 | 52.94 | 55.37 | 66.25 | 54.03 | 81.20 | 95.44 | 99.25 ± 0.15 |
| | FPR95 | 99.25 | 92.57 | 95.54 | 91.57 | 95.04 | 93.92 | 95.48 | 95.48 | 76.39 | 51.06 | 3.77 ± 0.92 |
| | ID ACC | 68.72 | 70.79 | 70.51 | 70.40 | 67.44 | 68.09 | 70.40 | 70.73 | 70.52 | 70.18 | 68.49 ± 0.13 |
| Twitch-RU | AUROC | 41.23 | 58.67 | 55.68 | 57.72 | 46.48 | 50.89 | 87.90 | 55.72 | 93.12 | 98.10 | 99.58 ± 0.06 |
| | AUPR | 56.06 | 72.58 | 66.42 | 66.68 | 62.11 | 65.14 | 89.05 | 70.18 | 95.36 | 98.74 | 99.78 ± 0.04 |
| | FPR95 | 95.01 | 93.98 | 90.13 | 87.57 | 95.62 | 99.93 | 43.95 | 95.07 | 30.72 | 5.59 | 1.14 ± 0.35 |
| | ID ACC | 68.72 | 70.79 | 70.51 | 70.40 | 67.44 | 68.09 | 70.40 | 70.73 | 70.52 | 70.18 | 68.49 ± 0.13 |

Table 12: Model performance on `Cora` with three types of OOD (**S**tructure manipulation, **F**eature interpolation, and **L**abel leave-out).

| Dataset | Metrics | Non-OOD Exposure | | | | | | | Real OOD Exposure | | | Ours |
| | | MSP | ODIN | Mahalanobis | Energy | GKDE | GPN | GNNSAFE | OE | Energy FT | GNNSAFE++ | GOLD |
|---|---|---|---|---|---|---|---|---|---|---|---|---|
| Cora-S | AUROC | 70.90 | 49.92 | 46.68 | 71.73 | 68.61 | 77.47 | 87.52 | 67.98 | 75.88 | 90.62 | 95.48 ± 0.28 |
| | AUPR | 45.73 | 27.01 | 29.03 | 46.08 | 44.26 | 53.26 | 77.46 | 46.93 | 49.18 | 81.88 | 91.06 ± 0.32 |
| | FPR95 | 87.30 | 100.00 | 98.19 | 88.74 | 84.34 | 76.22 | 73.15 | 95.31 | 67.73 | 53.51 | 21.86 ± 0.97 |
| | ID ACC | 75.50 | 74.90 | 74.90 | 76.00 | 73.70 | 76.50 | 75.80 | 71.80 | 75.50 | 76.10 | 77.4 ± 0.56 |
| Cora-F | AUROC | 85.39 | 49.88 | 49.93 | 86.15 | 82.79 | 85.88 | 93.44 | 81.83 | 88.15 | 95.56 | 96.64 ± 0.15 |
| | AUPR | 73.70 | 26.96 | 31.95 | 74.42 | 66.52 | 73.79 | 88.19 | 70.84 | 75.99 | 90.27 | 93.82 ± 0.24 |
| | FPR95 | 64.88 | 100.00 | 99.93 | 65.81 | 68.24 | 56.17 | 38.92 | 83.79 | 47.53 | 27.73 | 14.35 ± 2.05 |
| | ID ACC | 75.30 | 75.00 | 74.90 | 76.10 | 74.80 | 77.00 | 76.40 | 73.30 | 75.30 | 76.80 | 76.77 ± 0.21 |
| Cora-L | AUROC | 91.36 | 49.80 | 67.62 | 91.40 | 57.23 | 90.34 | 92.80 | 89.47 | 91.36 | 92.75 | 95.40 ± 0.17 |
| | AUPR | 78.03 | 24.27 | 42.31 | 78.14 | 27.50 | 82.21 | 82.21 | 77.01 | 78.49 | 82.64 | 88.65 ± 0.25 |
| | FPR95 | 34.99 | 100.00 | 90.77 | 41.08 | 88.95 | 37.42 | 30.83 | 46.55 | 37.83 | 34.08 | 17.28 ± 0.50 |
| | ID ACC | 88.92 | 88.92 | 88.92 | 88.92 | 89.87 | 91.46 | 88.92 | 87.97 | 90.51 | 91.46 | 90.82 ± 0.55 |

Table 13: Model performance on `Amazon` with three types of OOD (**S**tructure manipulation, **F**eature interpolation, and **L**abel leave-out).

| Dataset | Metrics | Non-OOD Exposure | | | | | | | Real OOD Exposure | | | Ours |
| | | MSP | ODIN | Mahalanobis | Energy | GKDE | GPN | GNNSAFE | OE | Energy FT | GNNSAFE++ | GOLD |
|---|---|---|---|---|---|---|---|---|---|---|---|---|
| Amazon-S | AUROC | 98.27 | 93.24 | 71.69 | 98.51 | 76.39 | 97.17 | 99.58 | 99.60 | 98.83 | 99.82 | 99.99 ± 0.03 |
| | AUPR | 98.54 | 95.26 | 79.01 | 98.72 | 81.58 | 96.39 | 99.76 | 99.61 | 99.14 | 99.89 | 99.99 ± 0.02 |
| | FPR95 | 6.13 | 65.44 | 99.91 | 4.97 | 99.25 | 11.65 | 0.00 | 0.51 | 1.31 | 0.00 | 0 ± 0 |
| | ID ACC | 92.84 | 92.84 | 92.79 | 92.86 | 87.57 | 88.51 | 92.61 | 92.61 | 92.79 | 92.22 | 92.03 ± 0.24 |
| Amazon-F | AUROC | 97.31 | 81.15 | 76.50 | 97.87 | 58.96 | 87.91 | 98.55 | 98.39 | 98.68 | 99.64 | 99.17 ± 0.02 |
| | AUPR | 95.16 | 78.47 | 71.14 | 95.64 | 66.76 | 84.77 | 98.99 | 96.24 | 96.82 | 99.68 | 99.31 ± 0.06 |
| | FPR95 | 8.72 | 100.0 | 76.12 | 6.00 | 99.28 | 49.11 | 0.31 | 4.34 | 2.84 | 0.13 | 0.14 ± 0.03 |
| | ID ACC | 92.89 | 92.71 | 92.86 | 92.96 | 86.18 | 90.05 | 92.30 | 92.30 | 92.52 | 92.39 | 91.76 ± 0.57 |
| Amazon-L | AUROC | 93.97 | 65.97 | 73.25 | 93.81 | 65.58 | 92.72 | 97.35 | 95.39 | 96.61 | 97.51 | 97.26 ± 0.27 |
| | AUPR | 91.32 | 57.80 | 66.89 | 91.13 | 65.20 | 90.34 | 97.12 | 92.53 | 94.92 | 97.07 | 97.46 ± 0.29 |
| | FPR95 | 26.65 | 90.23 | 74.30 | 28.48 | 96.87 | 37.16 | 6.59 | 17.72 | 13.78 | 6.18 | 6.06 ± 1.81 |
| | ID ACC | 95.76 | 96.08 | 95.76 | 95.72 | 89.37 | 90.07 | 95.76 | 95.72 | 94.83 | 95.84 | 95.18 ± 0.81 |

Table 14: Model performance on `Coauthor` with three types of OOD (**S**tructure manipulation, **F**eature interpolation, and **L**abel leave-out).

| Dataset | Metrics | Non-OOD Exposure | | | | | | | Real OOD Exposure | | | Ours |
|---|---|---|---|---|---|---|---|---|---|---|---|---|
| | | MSP | ODIN | Mahalanobis | Energy | GKDE | GPN | GNNSAFE | OE | Energy FT | GNNSAFE++ | GOLD |
| Coauthor-S | AUROC | 95.30 | 52.14 | 80.46 | 96.18 | 65.87 | 34.67 | 99.60 | 97.86 | 98.84 | 99.99 | 99.62 ± 0.02 |
| | AUPR | 94.37 | 48.83 | 76.65 | 95.25 | 72.65 | 40.21 | 99.69 | 96.81 | 97.78 | 99.99 | 99.78 ± 0.01 |
| | FPR95 | 24.75 | 99.92 | 70.75 | 18.02 | 99.48 | 99.57 | 0.26 | 9.23 | 3.97 | 0.02 | 0.01 ± 0.01 |
| | ID ACC | 92.47 | 92.34 | 92.33 | 92.75 | 88.62 | 89.45 | 92.73 | 92.60 | 92.61 | 92.92 | 91.41 ± 0.16 |
| Coauthor-F | AUROC | 97.05 | 51.54 | 93.23 | 97.88 | 80.69 | 81.77 | 99.64 | 99.04 | 99.43 | 99.97 | 99.78 ± 0.15 |
| | AUPR | 96.93 | 45.50 | 90.88 | 97.69 | 86.47 | 80.56 | 99.66 | 98.80 | 99.25 | 99.95 | 99.86 ± 0.09 |
| | FPR95 | 15.55 | 100.0 | 28.10 | 9.75 | 96.57 | 74.46 | 0.51 | 4.44 | 2.25 | 0.09 | 0.03 ± 0.01 |
| | ID ACC | 92.45 | 92.39 | 92.34 | 92.75 | 84.72 | 87.05 | 92.73 | 92.64 | 92.50 | 92.87 | 91.81 ± 0.26 |
| Coauthor-L | AUROC | 94.88 | 51.44 | 85.36 | 95.87 | 61.15 | 93.24 | 97.23 | 96.04 | 96.23 | 97.89 | 97.63 ± 0.16 |
| | AUPR | 97.99 | 74.79 | 93.61 | 98.34 | 81.39 | 97.55 | 98.98 | 98.50 | 98.51 | 99.24 | 99.06 ± 0.07 |
| | FPR95 | 23.81 | 100.0 | 45.41 | 18.69 | 94.60 | 34.78 | 12.06 | 18.17 | 17.07 | 9.43 | 9.46 ± 0.3 |
| | ID ACC | 95.18 | 95.15 | 95.19 | 95.20 | 89.05 | 91.68 | 95.21 | 95.10 | 95.20 | 95.24 | 94.84 ± 0.03 |

Table 15: Model performance on OOD datasets of paper published in 2018, 2019, and 2020 on `Arxiv`.

| Dataset | Metrics | Non-OOD Exposure | | | | | | | Real OOD Exposure | | | Ours |
|---|---|---|---|---|---|---|---|---|---|---|---|---|
| | | MSP | ODIN | Mahalanobis | Energy | GKDE | GPN | GNNSAFE | OE | Energy FT | GNNSAFE++ | GOLD |
| Arxiv-2018 | AUROC | 61.66 | 53.49 | 57.08 | 61.75 | 56.29 | OOM | 66.47 | 67.72 | 69.58 | 70.40 | 69.74 ± 0.28 |
| | AUPR | 70.63 | 63.06 | 65.09 | 70.41 | 66.78 | OOM | 74.99 | 75.74 | 76.31 | 78.62 | 77.12 ± 0.23 |
| | FPR95 | 91.67 | 100.0 | 93.69 | 91.74 | 94.31 | OOM | 89.44 | 86.67 | 82.10 | 81.47 | 83.20 ± 0.57 |
| | ID ACC | 53.78 | 51.39 | 51.59 | 53.36 | 50.76 | OOM | 53.39 | 52.39 | 53.26 | 53.50 | 50.59 ± 0.53 |
| Arxiv-2019 | AUROC | 63.07 | 53.95 | 56.76 | 63.16 | 57.87 | OOM | 68.36 | 69.33 | 70.58 | 72.16 | 72.46 ± 0.35 |
| | AUPR | 66.00 | 56.07 | 57.85 | 65.78 | 62.34 | OOM | 71.57 | 72.15 | 72.03 | 75.43 | 75.41 ± 0.38 |
| | FPR95 | 90.82 | 100.0 | 94.01 | 90.96 | 93.97 | OOM | 88.02 | 85.52 | 81.30 | 79.33 | 81.16 ± 0.58 |
| | ID ACC | 53.78 | 51.39 | 51.59 | 53.36 | 50.76 | OOM | 53.39 | 52.39 | 53.26 | 53.50 | 50.59 ± 0.53 |
| Arxiv-2020 | AUROC | 67.00 | 55.78 | 56.92 | 67.70 | 60.79 | OOM | 78.35 | 72.35 | 74.53 | 81.75 | 79.50 ± 0.11 |
| | AUPR | 90.92 | 87.41 | 85.95 | 91.15 | 88.74 | OOM | 94.76 | 92.57 | 93.08 | 95.57 | 95.02 ± 0.04 |
| | FPR95 | 89.28 | 100.0 | 95.01 | 89.69 | 93.31 | OOM | 83.57 | 83.28 | 78.36 | 71.50 | 77.36 ± 0.75 |
| | ID ACC | 53.78 | 51.39 | 51.59 | 53.36 | 50.76 | OOM | 53.39 | 52.39 | 53.26 | 53.50 | 50.59 ± 0.53 |

Table 16: Extended ablation performance of individual subsets.

| Dataset | Metrics | GNNSᴀꜰᴇ | GNNSᴀꜰᴇ++ | w/o Adv. | w/o Det. | GOLD |
|---|---|---|---|---|---|---|
| Twitch-ES | AUROC | 49.07 | 94.54 | 69.10 | 57.65 | 99.72 |
| | AUPR | 57.62 | 97.17 | 75.86 | 65.82 | 99.82 |
| | FPR95 | 93.98 | 44.06 | 85.82 | 91.65 | 0.44 |
| | ID ACC | 70.40 | 70.18 | 70.97 | 70.97 | 68.49 |
| Twitch-FR | AUROC | 63.49 | 93.45 | 93.86 | 88.98 | 99.08 |
| | AUPR | 66.25 | 95.44 | 95.45 | 92.61 | 99.25 |
| | FPR95 | 90.80 | 51.06 | 39.44 | 70.84 | 3.77 |
| | ID ACC | 70.40 | 70.18 | 70.97 | 70.97 | 68.49 |
| Twitch-RU | AUROC | 87.90 | 98.10 | 90.81 | 86.48 | 99.58 |
| | AUPR | 89.05 | 98.74 | 94.75 | 93.30 | 99.78 |
| | FPR95 | 43.95 | 5.59 | 53.87 | 77.01 | 1.14 |
| | ID ACC | 70.40 | 70.18 | 70.97 | 70.97 | 68.49 |
| Cora-S | AUROC | 87.52 | 90.62 | 90.05 | 93.33 | 95.48 |
| | AUPR | 77.46 | 81.88 | 83.04 | 87.13 | 91.06 |
| | FPR95 | 73.15 | 53.51 | 59.45 | 31.98 | 21.86 |
| | ID ACC | 75.80 | 76.10 | 67.70 | 75.60 | 77.40 |
| Cora-F | AUROC | 93.44 | 95.56 | 94.43 | 95.24 | 96.64 |
| | AUPR | 88.19 | 90.27 | 91.74 | 91.23 | 93.82 |
| | FPR95 | 38.92 | 27.73 | 29.54 | 26.74 | 14.35 |
| | ID ACC | 76.40 | 76.80 | 76.50 | 75.70 | 76.77 |
| Cora-L | AUROC | 89.47 | 92.75 | 84.45 | 91.71 | 95.40 |
| | AUPR | 82.21 | 82.64 | 65.90 | 81.98 | 88.65 |
| | FPR95 | 30.83 | 34.08 | 50.00 | 43.31 | 17.28 |
| | ID ACC | 88.92 | 91.46 | 88.60 | 90.80 | 90.82 |
| Arxiv-2018 | AUROC | 66.47 | 70.40 | 64.97 | 65.55 | 69.74 |
| | AUPR | 74.99 | 78.62 | 72.71 | 73.63 | 77.12 |
| | FPR95 | 89.44 | 81.47 | 90.12 | 91.19 | 83.20 |
| | ID ACC | 53.39 | 53.50 | 49.89 | 49.66 | 50.59 |
| Arxiv-2019 | AUROC | 68.36 | 72.16 | 67.21 | 67.13 | 72.46 |
| | AUPR | 71.57 | 75.43 | 69.70 | 69.06 | 75.41 |
| | FPR95 | 88.02 | 79.33 | 88.97 | 90.27 | 81.16 |
| | ID ACC | 53.39 | 53.50 | 49.89 | 49.66 | 50.59 |
| Arxiv-2020 | AUROC | 78.35 | 81.75 | 77.11 | 77.04 | 79.50 |
| | AUPR | 94.76 | 95.57 | 94.39 | 94.45 | 95.02 |
| | FPR95 | 83.57 | 71.50 | 85.40 | 87.54 | 77.36 |
| | ID ACC | 53.39 | 53.50 | 49.89 | 49.66 | 50.59 |

Table 17: Extended adversarial training effectiveness analysis of individual subsets.

| Dataset | Metrics | GNNSafe++ | Dif. Once | Dif. Multi | Real OOD | GOLD |
|---|---|---|---|---|---|---|
| Twitch-ES | AUROC | 94.54 | 69.10 | 66.52 | 98.99 | 99.72 |
|  | AUPR | 97.17 | 75.86 | 73.49 | 99.52 | 99.82 |
|  | FPR95 | 44.06 | 85.82 | 87.07 | 1.38 | 0.44 |
|  | ID ACC | 70.18 | 70.40 | 71.12 | 70.45 | 68.49 |
| Twitch-FR | AUROC | 93.45 | 93.86 | 95.20 | 94.51 | 99.08 |
|  | AUPR | 95.44 | 95.45 | 96.54 | 96.33 | 99.25 |
|  | FPR95 | 51.06 | 39.44 | 31.08 | 40.62 | 3.77 |
|  | ID ACC | 70.18 | 70.40 | 71.12 | 70.45 | 68.49 |
| Twitch-RU | AUROC | 98.10 | 90.81 | 91.28 | 99.24 | 99.58 |
|  | AUPR | 98.74 | 94.75 | 95.10 | 99.65 | 99.78 |
|  | FPR95 | 5.59 | 53.87 | 82.84 | 1.16 | 1.14 |
|  | ID ACC | 70.18 | 70.40 | 71.12 | 70.45 | 68.49 |
| Cora-S | AUROC | 90.62 | 90.05 | 95.69 | 94.12 | 95.48 |
|  | AUPR | 81.88 | 83.04 | 91.59 | 89.91 | 91.06 |
|  | FPR95 | 53.51 | 59.45 | 22.30 | 37.11 | 21.86 |
|  | ID ACC | 76.10 | 67.70 | 76.10 | 75.90 | 77.40 |
| Cora-F | AUROC | 95.56 | 94.43 | 96.02 | 97.60 | 96.64 |
|  | AUPR | 90.27 | 91.74 | 92.99 | 94.23 | 93.82 |
|  | FPR95 | 27.73 | 29.54 | 18.94 | 10.27 | 14.35 |
|  | ID ACC | 76.80 | 76.50 | 77.30 | 71.70 | 76.77 |
| Cora-L | AUROC | 92.75 | 84.45 | 86.76 | 95.04 | 95.40 |
|  | AUPR | 82.64 | 65.90 | 70.70 | 86.01 | 88.65 |
|  | FPR95 | 34.08 | 50.00 | 49.09 | 17.24 | 17.28 |
|  | ID ACC | 91.46 | 88.60 | 88.29 | 87.65 | 90.82 |
| Arxiv-2018 | AUROC | 70.40 | 64.97 | 67.26 | 75.32 | 69.74 |
|  | AUPR | 78.62 | 72.71 | 74.72 | 80.89 | 77.12 |
|  | FPR95 | 81.47 | 90.12 | 85.23 | 72.40 | 83.20 |
|  | ID ACC | 53.50 | 53.39 | 50.77 | 49.99 | 50.59 |
| Arxiv-2019 | AUROC | 72.16 | 67.21 | 69.66 | 77.98 | 72.46 |
|  | AUPR | 75.43 | 69.70 | 72.05 | 79.56 | 75.41 |
|  | FPR95 | 79.33 | 88.97 | 83.33 | 95.92 | 81.16 |
|  | ID ACC | 53.50 | 53.39 | 50.77 | 49.99 | 50.59 |
| Arxiv-2020 | AUROC | 81.75 | 77.11 | 79.52 | 83.41 | 79.50 |
|  | AUPR | 95.57 | 94.39 | 94.95 | 95.92 | 95.02 |
|  | FPR95 | 71.50 | 85.40 | 77.52 | 64.93 | 77.36 |
|  | ID ACC | 53.50 | 53.39 | 50.77 | 49.99 | 50.59 |

Table 18: Extended energy regulariser effectiveness analysis on `Twitch`.

| $\mathcal{L}_{\text{Unc}}$ | $\mathcal{L}_{\text{EReg}}$ | $\mathcal{L}_{\text{DReg}}$ | Twitch-ES | | | | Twitch-FR | | | | Twitch-RU | | | |
|---|---|---|---|---|---|---|---|---|---|---|---|---|---|---|
|  |  |  | AUROC | AUPR | FPR | ID Acc | AUROC | AUPR | FPR | ID Acc | AUROC | AUPR | FPR | ID Acc |
|  |  |  | 74.33 | 76.23 | 52.97 | 68.97 | 98.34 | 98.72 | 2.58 | 68.97 | 26.92 | 48.67 | 91.70 | 68.97 |
| ✓ |  |  | 17.39 | 44.31 | 98.04 | 70.15 | 5.69 | 34.24 | 99.07 | 70.15 | 7.48 | 43.30 | 96.42 | 70.15 |
|  | ✓ |  | 60.03 | 68.63 | 92.36 | 70.98 | 90.21 | 92.64 | 66.57 | 70.98 | 83.81 | 88.85 | 77.77 | 70.98 |
|  |  | ✓ | 18.69 | 44.81 | 97.89 | 70.79 | 96.40 | 94.65 | 8.85 | 70.79 | 92.04 | 91.19 | 26.89 | 70.79 |
| ✓ | ✓ |  | 59.40 | 68.20 | 92.77 | 70.99 | 91.91 | 93.23 | 56.07 | 70.99 0 | 79.32 | 83.03 | 79.59 | 70.99 |
| ✓ |  | ✓ | 7.50 | 42.10 | 99.07 | 70.90 | 99.04 | 98.15 | 1.30 | 70.90 | 86.76 | 86.13 | 37.49 | 70.90 |
|  | ✓ | ✓ | 90.55 | 94.20 | 41.98 | 69.64 | 88.95 | 91.30 | 45.86 | 69.64 | 90.35 | 80.94 | 42.09 | 69.64 |
| GOLD | | | 99.72 | 99.82 | 0.44 | 68.49 | 99.08 | 99.25 | 3.77 | 68.49 | 99.58 | 99.78 | 1.14 | 68.49 |

Table 19: Extended energy regulariser effectiveness analysis on `Cora`.

| $\mathcal{L}_{\text{Unc}}$ | $\mathcal{L}_{\text{EReg}}$ | $\mathcal{L}_{\text{DReg}}$ | Cora-S | | | | Cora-F | | | | Cora-L | | | |
|---|---|---|---|---|---|---|---|---|---|---|---|---|---|---|
| | | | AUROC | AUPR | FPR | ID Acc | AUROC | AUPR | FPR | ID Acc | AUROC | AUPR | FPR | ID Acc |
| | | | 86.44 | 91.26 | 35.20 | 67.70 | 13.00 | 16.79 | 99.19 | 70.80 | 83.98 | 76.01 | 90.06 | 90.19 |
| ✓ | | | 68.06 | 67.14 | 89.18 | 80.00 | 68.01 | 77.40 | 45.70 | 75.80 | 76.20 | 60.36 | 96.65 | 87.34 |
| | ✓ | | 94.70 | 90.09 | 30.54 | 74.30 | 10.41 | 15.72 | 100 | 76.20 | 92.90 | 84.14 | 31.64 | 90.82 |
| | | ✓ | 89.18 | 80.96 | 65.18 | 72.80 | 79.83 | 74.36 | 90.69 | 70.00 | 84.16 | 68.39 | 49.59 | 85.44 |
| ✓ | ✓ | | 47.95 | 49.68 | 91.32 | 77.60 | 47.63 | 54.31 | 99.41 | 76.60 | 8.30 | 13.82 | 99.80 | 89.55 |
| ✓ | | ✓ | 86.26 | 76.48 | 64.59 | 69.70 | 79.40 | 73.66 | 93.65 | 67.50 | 86.43 | 71.52 | 62.37 | 86.39 |
| | ✓ | ✓ | 93.94 | 88.86 | 28.99 | 75.60 | 95.54 | 92.69 | 22.05 | 76.70 | 90.35 | 80.94 | 42.09 | 87.34 |
| | GOLD | | 95.48 | 91.06 | 21.86 | 77.40 | 96.64 | 93.82 | 14.35 | 76.77 | 95.40 | 88.65 | 17.28 | 90.82 |

Table 20: Extended energy regulariser effectiveness analysis on `Amazon`.

| $\mathcal{L}_{\text{Unc}}$ | $\mathcal{L}_{\text{EReg}}$ | $\mathcal{L}_{\text{DReg}}$ | Amazon-S | | | | Amazon-F | | | | Amazon-L | | | |
|---|---|---|---|---|---|---|---|---|---|---|---|---|---|---|
| | | | AUROC | AUPR | FPR | ID Acc | AUROC | AUPR | FPR | ID Acc | AUROC | AUPR | FPR | ID Acc |
| | | | 96.67 | 97.48 | 20.16 | 88.35 | 1.21 | 26.70 | 99.35 | 92.11 | 94.64 | 93.84 | 20.66 | 95.76 |
| ✓ | | | 84.90 | 90.80 | 100.00 | 92.17 | 21.53 | 30.92 | 86.18 | 92.96 | 95.17 | 93.45 | 18.21 | 95.96 |
| | ✓ | | 52.82 | 69.21 | 100.00 | 92.74 | 1.58 | 26.81 | 98.47 | 91.89 | 91.53 | 89.77 | 35.26 | 95.72 |
| | | ✓ | 100.00 | 100.00 | 0.00 | 91.35 | 99.76 | 99.55 | 0.63 | 91.62 | 93.39 | 90.94 | 24.20 | 95.39 |
| ✓ | ✓ | | 85.72 | 91.06 | 99.97 | 92.48 | 57.99 | 44.03 | 55.73 | 92.63 | 69.74 | 56.45 | 68.85 | 94.79 |
| ✓ | | ✓ | 100.00 | 100.00 | 0.00 | 92.33 | 98.61 | 99.09 | 0.29 | 91.76 | 95.11 | 91.99 | 13.34 | 95.52 |
| | ✓ | ✓ | 98.58 | 99.21 | 0.00 | 92.25 | 98.36 | 98.75 | 0.60 | 92.04 | 97.13 | 97.29 | 9.61 | 94.14 |
| | GOLD | | 99.98 | 99.99 | 0.00 | 92.03 | 99.17 | 99.31 | 0.14 | 91.76 | 97.26 | 97.46 | 6.06 | 95.18 |

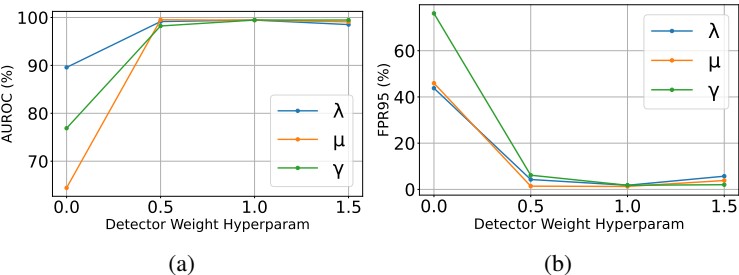

(a)                                          (b)

Figure 6: The `Twitch` dataset was utilised for conducting hyper-parameter sensitivity analysis. (6a) and (6b) are Hyper-parameter sensitivity of different weights in Eq. 13 for Detector measured by AUROC and FPR95.

## A.10 Ablation Study Visualisation

To explore how the different modules contribute to OOD detection, we present further energy distribution visualisations in Figure 7. The adversarial training can help to maintain the closeness between synthetic and ID data, preventing the synthetic samples from diverging too far from real ID/OOD data to bias the detector. Our method alternates between two tasks: (1) the latent diffusion model pulls latent embeddings of ID data and generated pseudo-OOD embeddings closer, while (2) the GNN & detector push their energies apart. Without this component, the pseudo-OOD distribution diverges significantly compared to GOLD in Figure 7 c/f, where synthetic embeddings are regularly updated. This divergence makes OOD data indistinct from ID data, leading to poor performance in the ablation study. The detector can decrease the overlap of energy distribution between the ID and OOD samples, leading to better energy-based OOD detection. Figure 7 b/e shows that w/o detector will lead to a large overlap of energy distribution between ID and OOD samples. This overlap occurs because the energy scores, derived from prediction logits of the GNN classifier, become indistinct as the number of predicted classes increases and when the classifier struggles to distinguish certain classes. Thus, introducing a dedicated detector to further discern energy scores enhances detection by reducing the number of output classes.

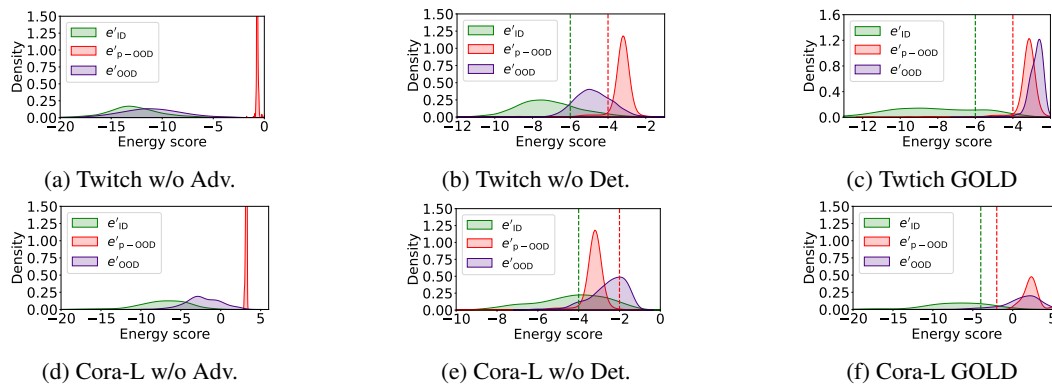

Figure 7: **Visualisation of the energy score distributions of GOLD without adversarial training or the use of detector for `Twitch` and `Cora-L` datasets.** (a) and (d) illustrate the energy score gaps w/o adversarial training, where the energy of p-OOD data will be diverged too far and fail to diverge the energy of real OOD. (b) and (e) shows the energy scores derived from the GNN classifier without our proposed detector, where the energy scores cannot be effectively separated. (c) and (f) demonstrates the ability of GOLD to effectively distinguish the ID and OOD energy distributions, illustrating the effectiveness of the adversarial and detector components.

## A.11    LATENT GENERATIVE MODEL

In this section, we provide description of the two latent generative models utilised: The variational autoencoder and the latent diffusion model.

### A.11.1    VARIATIONAL AUTOENCODER

The variational autoencoder (VAE) is a generative model consisting of an encoder that learns latent variables from training data and a decoder that then uses those latent variables to reconstruct the input data. VAEs are trained to optimise a lower bound on the marginal log-likelihood $\log p_\theta(x)$ over the data by using a learned approximate posterior $q_\phi(h|x)$, as follows:

$$\mathcal{L}(\theta, \phi; x) = \mathbb{E}_{q_\phi(h|x)}[\log p_\theta(x|h)] - D_{KL}(q_\phi(h|x)||p(h))$$

s.t the first term is the reconstruction loss, and the second term is the KL divergence of the approximate from the true posterior. The trained approximate posterior $q_\phi(h|x)$ would thus act as an encoder that maps the data $x$ to a lower dimensional latent representation, and latent samples $h$ can be drawn via the reparametrisation trick:

$$h = \mu_\phi(x) + \sigma_\phi(x) \odot \epsilon, \text{ where } \epsilon \sim \mathcal{N}(0, I) \text{ if the models are Gaussian.}$$

We set the encoder hidden dimension size to be 512, the decoder dimension to be 256, and layer sizes to be 2.

### A.11.2    LATENT DIFFUSION MODEL

The latent diffusion model consists of a forward diffusion and a backward denoising process on a set of latent representations (Ho et al., 2020; Zhou et al., 2024; Evdaimon et al., 2024). In our GOLD, a latent node representation $\mathbf{h}_0 \in \mathbb{R}^{d'}$ is initialised at timestep 0 from the GNN encodings $\mathbf{H}$ in Eq. 7. At the forward process, the model progressively adds Gaussian noise to the latent node representation $\mathbf{h}_0$, according to a known variance schedule $\beta_1, \cdots, \beta_T$, for $0 < \beta_1 < ...\beta_T < 1$. This process will produce a sequence of increasingly noisy vectors $(\mathbf{h}_1, \cdots \mathbf{h}_T)$ with timestep $t = \{1, 2, 3, \ldots, T\}$. Denoting $a_t = 1 - \beta_t$ and $\bar{a}_t = \prod_{i=1}^{t} a_i$, we can derive a closed form for obtaining the representation at any timestep $t$ given the initial representation $\mathbf{h}_0$:

$$\mathbf{h}_t \sim \mathcal{N}\left(\sqrt{\bar{a}_t}\mathbf{h}_0, (1 - \bar{a}_t)\mathbf{I}\right). \tag{20}$$

The backward denoising process involves predicting the noise added to the representation at a given timestep via a denoising model $D$ (e.g., MLP). To train the latent diffusion model, we minimise the

mean squared error loss between the added noise $\epsilon \sim \mathcal{N}(\mathbf{0}, \mathbf{I})$ and the predicted noise from the noisy representation $\mathbf{h}_t$ at a given timestep $t$ with the reparameterisation trick:

$$\min_D \mathcal{L}_{\text{Gen}}, \text{ where } \mathcal{L}_{\text{Gen}} = \mathbb{E}_{\mathbf{h}_0, \epsilon, t} \left[ \left\| \epsilon - D\left(\sqrt{\bar{a}_t}\mathbf{h}_0 + \sqrt{(1 - \bar{a}_t)}\epsilon, t\right) \right\|_2^2 \right]. \tag{21}$$

| Hyperparams | | AUROC | AUPR | FPR | ID ACC |
|---|---|---|---|---|---|
| $\beta_1$ | 0.00001 | 99.43 | 99.59 | 1.77 | 68.48 |
| | **0.0001** | 99.46 | 99.62 | 1.78 | 68.49 |
| | 0.001 | 99.52 | 99.66 | 1.44 | 68.11 |
| | 0.01 | 99.46 | 99.60 | 1.99 | 67.71 |
| $\beta_T$ | 0.005 | 85.91 | 88.79 | 44.77 | 71.04 |
| | **0.02** | 99.46 | 99.62 | 1.78 | 68.49 |
| | 0.1 | 82.02 | 86.79 | 55.27 | 68.40 |
| $T$ | 400 | 96.59 | 97.84 | 18.23 | 69.43 |
| | 500 | 95.15 | 96.38 | 21.32 | 68.99 |
| | **600** | 99.46 | 99.62 | 1.78 | 68.49 |
| | 700 | 98.80 | 99.27 | 4.07 | 68.09 |
| | 800 | 92.02 | 95.68 | 66.94 | 68.85 |
| | 1000 | 17.35 | 45.22 | 99.57 | 71.20 |

Table 21: Performance comparison for different hyperparameters for Diffusion model on Twitch dataset. Default values are highlighted in **Bold**.

We set the diffusion model parameter $\beta$ to be a sequence of linearly increasing constants from $\beta_1 = 10^{-4}$ to $\beta_T = 0.02$ as presented in (Ho et al., 2020; Rombach et al., 2022). A hyperparameter sensitivity experiment on the Twitch dataset is provided in Table 21. Generally, a larger (smaller) $\beta$ adds or removes more (less) noise at each step. A larger $T$ increases noise corruption, making recovery harder but with more output variation, while a smaller $T$ reduces noise corruption, making recovery easier but limiting variation. $\beta_1$ typically does not affect performance, while $\beta_T$ is more sensitive, reflecting higher/lower corruption at the end of the timestep. The value of $T$ also impacts performance, with non-default values either limiting or excessively diversifying the synthetic samples.

## A.12 COMPUTATIONAL COST

In this section, we provide the computational cost of GOLD against SOTA baselines. GOLD outperforms the baselines with a rough trade-off of 2x training time and memory usage.

Table 22: **Computation cost (one 32GB (32768MiB) NVIDIA V100 GPU) and OOD detection performance of GOLD (Non-OOD Exposed) against both Non- and Real-OOD exposed SOTA baselines.** The 'Train' column is the training convergence time in seconds. The 'Test' column is the inference time in seconds. The 'Mem.' column is the maximum memory usage in Mebibytes (MiB). The 'FPR95' column is the OOD detection performance in %, the lower the better. **The inference time of these methods is the same with the same backbone GNN.**

| | Twitch | | | | Cora-F | | | | Amazon-F | | | | Coauthor-F | | | | Arxiv | | | |
|---|---|---|---|---|---|---|---|---|---|---|---|---|---|---|---|---|---|---|---|---|
| | Train | Test | Mem. | FPR95($\downarrow$) | Train | Test | Mem. | FPR95($\downarrow$) | Train | Test | Mem. | FPR95($\downarrow$) | Train | Test | Mem. | FPR95($\downarrow$) | Train | Test | Mem. | FPR95($\downarrow$) |
| GNNSAFE (Non) | 2.41 | 0.08 | 667 | 76.24 | 4.40 | 0.03 | 465 | 38.92 | 13.51 | 0.04 | 665 | 0.31 | 57.80 | 0.35 | 1523 | 0.51 | 85.23 | 0.40 | 3370 | 87.01 |
| GNNSAFE++ (Real) | 4.74 | 0.09 | 667 | 33.57 | 5.32 | 0.03 | 465 | 27.73 | 18.40 | 0.05 | 665 | 0.13 | 67.83 | 0.36 | 1523 | 0.09 | 132.36 | 0.40 | 3370 | 77.43 |
| GOLD w/ VAE (Non) | 2.78 | 0.09 | 1427 | 3.03 | 3.91 | 0.04 | 1081 | 23.60 | 12.52 | 0.05 | 1319 | 0.15 | 55.65 | 0.35 | 2439 | 0.23 | 80.77 | 0.45 | 9039 | 81.95 |
| GOLD w/ LDM (Non) | 8.96 | 0.10 | 1452 | 1.71 | 5.93 | 0.04 | 1083 | 14.51 | 39.04 | 0.07 | 1347 | 0.11 | 89.74 | 0.37 | 2515 | 0.01 | 244.95 | 0.47 | 10579 | 80.35 |

## A.13 ABLATION WITH ADDITIONAL BACKBONE

We provide the following experiments with two additional backbones: GAT (Velickovic et al., 2018) and MixHop (Abu-El-Haija et al., 2019). We compare these architectures against GNNSafe and NodeSafe and their OOD-exposed variants. To ensure a fair comparison, we maintain the same configuration as the original GCN implementation, with a hidden dimension of 64, two layers, 8 attention heads for GAT, and two hops for MixHop. The results shown in Table 23, demonstrate that GOLD outperforms other methods across the evaluated backbones.

Table 23: Ablation of different backbones

| Dataset | Backbone | Metrics | GNNSAFE | GNNSAFE++ | NODESAFE | NODESAFE++ | GOLD |
|---------|----------|---------|---------|-----------|----------|------------|------|
| Twitch | MixHop | AUROC | 72.08 | 95.07 | 57.91 | 95.08 | **96.94** |
| | | FPR95 | 73.70 | 33.46 | 93.76 | 30.71 | **17.98** |
| | | ID Acc | 69.66 | 66.04 | 70.09 | 70.56 | 67.58 |
| | GAT | AUROC | 83.08 | 97.51 | 54.78 | 95.07 | **98.64** |
| | | FPR95 | 50.46 | 20.43 | 93.24 | 30.71 | **1.42** |
| | | ID Acc | 68.21 | 68.54 | 68.40 | 70.56 | 67.32 |
| Cora | MixHop | AUROC | 88.65 | 91.33 | 82.60 | **92.79** | 91.42 |
| | | FPR95 | 59.08 | 44.59 | 60.22 | 38.63 | **25.09** |
| | | ID Acc | 79.52 | 80.66 | 82.16 | 81.45 | 80.67 |
| | GAT | AUROC | 91.62 | 92.50 | 85.55 | 92.32 | **94.66** |
| | | FPR95 | 33.81 | 33.44 | 55.20 | 34.93 | **19.63** |
| | | ID Acc | 79.44 | 79.52 | 81.06 | 80.23 | 78.40 |

