# OpenReview forum: "GOLD: Graph Out-of-Distribution Detection via Implicit Adversarial Latent Generation"
_ICLR.cc/2025/Conference — ICLR 2025 Spotlight_

### Official Review · Reviewer_yx8w · 2024-10-31

**Soundness:** 4
**Presentation:** 4
**Contribution:** 4
**Rating:** 8
**Confidence:** 3

**Summary:**

This paper proposes the GOLD, a novel framework for graph OOD detection that employs implicit adversarial learning to generate synthetic OOD instances without the need for pre-trained models or additional OOD data. By utilizing a latent generative model to produce embeddings mimicking in-distribution data, which are then differentiated from actual in-distribution embeddings by a graph neural network encoder and an OOD detector, the method is able to effectively simulate the OOD exposure. The framework is evaluated on five benchmark graph datasets, demonstrating superior OOD detection performance without using real OOD data compared with the state-of-the-art OOD exposure and non-exposure baselines.

**Strengths:**

1. The idea of implicit adversarial learning is novel.
2. The experiments are solid and comprehensive, especially the improvements achieved on the FPR95 dataset is impressive.
3. The paper is easy to follow and well-structured. The writing is good. The theoritical proofs are provided.

**Weaknesses:**

I didn't see obvious weaknesses. Here I provide some suggestions:

Could you provide a more in-depth analysis of the performance achievements on the FPR95 dataset, including an analysis from the perspective of the dataset's characteristics? Additionally, could you explain why similar significant improvements were not observed on other datasets (although the improvements on other datasets are also quite good)?

**Questions:**

See weakness.

---

> ### Author Response · Authors · 2024-11-19
> **Response to Reviewer yx8w (1/1)**
>
> We would like to express our sincere gratitude for your recognition of our work and your insightful suggestions. We have carefully addressed your comments as follows.
>
> > **Re W1:** Performance analysis
>
> For our model evaluation, we use several widely adopted metrics to assess OOD detection performance: **AUROC, AUPR, and FPR95**. AUROC and AUPR measure the trade-off between true positive rates (TPR) and false positive rates at various thresholds, while FPR95 focuses on performance under high-sensitivity conditions—specifically, the rate of misclassifying ID samples as OOD when the TPR is at 95%. Thus, **FPR95 provides a more noticeable score gain** under stricter criteria, reflecting improvements in detection performance.
>
> Moreover, with respect to the **dataset characteristics**, considering the multi-graph Twitch dataset, which consists of 3 real OOD graphs. As shown in Table 1 of the manuscript, the Non(Real)-OOD Exposure baselines GNNSafe(++) achieves an average FPR95 of 76.24% (33.57%), while our GOLD improves this to 1.78%. This improvement is likely because the **real OOD graphs exhibit a distribution similar to the training ID data**, making it difficult for Non-OOD exposed GNNSafe to distinguish between ID and OOD. In contrast, GNNSafe++ benefits from additional OOD training data, helping it better differentiate between ID and test OOD samples.
>
> A closer look at individual OOD test graphs performance (see Table 11 in Appendix A.9 and below) reveals that the **baselines perform significantly better on one particular OOD test graph**, Twitch-RU. This suggests that the higher-performing OOD graph has a greater distributional divergence from the ID data, while the lower-performing OOD graphs are more similar to the ID data. Notably, GNNSafe++ appears to strongly benefit from the exposure of an OOD graph that may be distributionally similar to Twitch-RU, boosting detection performance.
>
> | OOD Test data | Metric (%) | GNNSafe | GNNSafe++ | GOLD |
> |--|--|--|--|--|
> | Twitch-ES | AUROC | 49.07 | 94.54 | 99.72 |
> |           | FPR95 | 93.98 | 44.06 | **0.44**|
> | Twitch-FR | AUROC | 63.49 | 93.45 | 99.08 |
> |           | FPR95 | 90.80 | 51.06 | **3.77**|
> | Twitch-RU | AUROC | 87.90 | 98.10 |99.58 |
> |           | FPR95 | 43.95 | 5.59  | **1.14**|
>
> However, this may introduce bias, as the model’s performance could vary depending on the similarity between the OOD test data and the OOD training data. **Our proposed model addresses this bias** by simulating pseudo-OOD instances directly from the ID data and incrementally diversifying them to improve detection across a broader range of OOD distributions. Thereby, alleviating potential bias from the exposed OOD, greatly improving performance across diverse OOD test sets.

---

> > ### Comment · Reviewer_yx8w · 2024-11-25
> >
> > Dear Authors,
> >
> > Thank you for the clarifications and the responses. I am going to keep my acceptance score unchanged. Good luck!

---

> ### Author Response · Authors · 2024-11-25
>
> Dear Reviewer yx8w,
>
> Thank you once again for your recognition and thoughtful feedback on our submission. We will ensure the clarified information goes to the future version of the paper.

---

### Official Review · Reviewer_AALT · 2024-11-02

**Soundness:** 3
**Presentation:** 3
**Contribution:** 3
**Rating:** 8
**Confidence:** 3

**Summary:**

The paper presents GOLD, a novel framework aimed at enhancing Out-of-Distribution (OOD) detection in graph neural networks (GNNs) without relying on external OOD data. GOLD introduces an implicit adversarial training pipeline where a latent generative model (LGM) is trained to generate embeddings that mimic in-distribution (ID) data, while an OOD detector is optimized to increase divergence between ID embeddings and these synthetic pseudo-OOD embeddings. It effectively simulates OOD exposure, helping the model distinguish OOD nodes in graph data. Extensive experiments demonstrate that GOLD outperforms state-of-the-art methods in OOD detection across several benchmark datasets without the need for real OOD data.

**Strengths:**

- GOLD’s adversarial latent generation is a novel approach that synthesizes pseudo-OOD data without auxiliary datasets, making it efficient and broadly applicable.

- GOLD’s effectiveness is demonstrated through comprehensive experiments on five benchmark datasets, showing its robustness across various graph types and OOD scenarios.

-  The implicit adversarial objective and energy-based detection approach lead to a clear divergence between ID and OOD embeddings, validated by experimental visualizations.

- This paper is well-structured.

**Weaknesses:**

This paper has no obvious weaknesses except for the training computational cost induced by the pseudo-OOD data. However, this cost increase is acceptable. And the authors have also discusses this issue in POTENTIAL LIMITATIONS section.

**Questions:**

1. What does subscript (i.e., [0], [1]) in Eq(10) mean?

2. Have the authors tried other backbone models?

---

> ### Author Response · Authors · 2024-11-19
> **Response to Reviewer AALT (1/1)**
>
> We greatly appreciate your comments, as they are helpful in improving our work. We have thoroughly reviewed your questions and have addressed as below.
>
> > **Re W1:** Computational cost
>
> We would like to thank the reviewer for the careful attention to detail. As noted, we acknowledge the inherent training cost associated with using latent diffusion models. Demonstrated in Tables 1 & 5 of the manuscript, GOLD is able to achieve SOTA OOD detection performance and also delivers inference times that match current baselines. Given these improvements, we believe the trade-off between increased cost and enhanced performance is justified. We also appreciate the reviewer’s recognition that this cost increase is acceptable.
>
> Additionally, to further improve efficiency, we have experimented with an additional lightweight variant by using a latent VAE to replace the latent diffusion. This approach also delivers competitive performance, which demonstrates the efficacy of our method. Importantly, we would like to emphasise that our model maintains (almost) the same inference speed as the baseline methods, ensuring that the performance gains do not come at the expense of test-time efficiency.
>
> > **Re Q1:** Notation clarification in Eq. 10)
>
> Apologies for the unclear notation, the subscript represents the label of the corresponding logit value from the binary classifier (MLP) model after applying the Softmax function (i.e., [0] represents the ID class 0 and [1] represents the OOD class 1).
>
> We have highlighted the revision in Section 3.2 of the manuscript.
>
> > **Re Q2:** Different backbones
>
> We provide the following **experiments with two additional backbones: GAT \[1\] and MixHop \[2\]**, following GNNSafe and NodeSafe, where GAT and MixHop achieve second and third places, while GCN achieves the highest performance similar to their ablation studies. We compare these architectures against GNNSafe and NodeSafe, as well as their OOD-exposed variants. To ensure a fair comparison, we **maintain the same configuration as the original GCN implementation**, with a hidden dimension of 64, two layers, 8 attention heads for GAT, and two hops for MixHop. The results, shown below, demonstrate that GOLD outperforms other methods across the evaluated backbones.
> | **Dataset**         | **Backbone** | **Metrics** | **GNNSafe**                   | **GNNSafe++**     | **NodeSafe**   | **NodeSafe++**          | **GOLD**                       |
> |----------------------|--------------|-------------|-------------------------------|-------------------|----------------|--------------------------|--------------------------------|
> | **Twitch**           | MixHop       | AUROC       | $\color{red}{72.08}$      | 95.07             | 57.91          | $\underline{95.08}$      | $\color{teal}{\mathbf{96.94}}$ |
> |                      |              | FPR95       | $\color{red}{73.70}$      | 33.46             | 93.76          | $\underline{30.71}$      | $\color{teal}{\mathbf{17.98}}$ |
> |                      |              | ID Acc      | 69.66                         | 66.04             | 70.09          | 70.56                    | 67.58                          |
> |                      | GAT          | AUROC       | $\color{red}{83.08}$      | $\underline{97.51}$ | 54.78          | 96.98                   | $\color{teal}{\mathbf{98.64}}$ |
> |                      |              | FPR95       | $\color{red}{50.46}$      | 20.43 | 93.24          | $\underline{8.61}$                    | $\color{teal}{\mathbf{1.42}}$  |
> |                      |              | ID Acc      | 68.21                         | 68.54             | 68.40          | 67.73                    | 67.32                          |
> | **Cora**             | MixHop       | AUROC       | $\color{red}{88.65}$      | 91.33             | 82.60          | $\mathbf{92.79}$         | $\underline{\color{teal}{91.42}}$ |
> |                      |              | FPR95       | $\color{red}{59.08}$      | 44.59             | 60.22          | $\underline{38.63}$      | $\color{teal}{\mathbf{25.09}}$ |
> |                      |              | ID Acc      | 79.52                         | 80.66             | 82.16          | 81.45                    | 80.67                          |
> |                      | GAT          | AUROC       | $\color{red}{91.62}$      | $\underline{92.50}$ | 85.55          | 92.32                    | $\color{teal}{\mathbf{94.66}}$ |
> |                      |              | FPR95       | $\color{red}{33.81}$      | $\underline{33.44}$ | 55.20          | 34.93                    | $\color{teal}{\mathbf{19.63}}$ |
> |   |              | ID Acc      | 79.44 | 79.52  | 81.06          | 80.23                    | 78.40   |
>
> We have also updated this as highlighted in Appendix A.13 of the manuscript.
>
> [1] Veličković et.al., Graph Attention Networks. ICLR18.
>
> [2] Abu-El-Haija et.al., MixHop: Higher-Order Graph Convolutional Architectures via Sparsified Neighborhood Mixing. ICML19.

---

> > ### Comment · Reviewer_AALT · 2024-11-26
> >
> > Thanks for the reply. I have no more questions.

---

> > > ### Author Response · Authors · 2024-11-26
> > >
> > > Dear Reviewer AALT
> > >
> > > We sincerely appreciate your constructive feedback and your recognition of our work. We will make sure the clarification and additional experiments are updated in a future version of the manuscript.

---

### Official Review · Reviewer_SRXq · 2024-11-03

**Soundness:** 3
**Presentation:** 3
**Contribution:** 3
**Rating:** 8
**Confidence:** 3

**Summary:**

This paper introduces a new framework called GOLD, which aims to address the challenges faced by graph neural networks (GNNs) when dealing with out-of-distribution (OOD) test instances. The GOLD framework detects OOD nodes in graph data through an implicit adversarial learning process without relying on pre-trained models. The core of the framework is the alternating optimization framework.

**Strengths:**

1. The paper presents a method with notable novelty, particularly in how it handles the generation of OOD (Out-of-Distribution) samples based on ID (In-Distribution) data. This approach demonstrates creative problem-solving and provides a potentially valuable contribution to OOD detection research.
2. The experimental section is comprehensive, covering multiple datasets and providing a range of performance metrics. This thorough evaluation supports the robustness of the method and suggests that it may perform effectively across diverse scenarios.

**Weaknesses:**

1. Certain issues in the manuscript reduce its overall clarity and precision. For example, Figure 1(d) appears to be blank, which may hinder understanding and interpretation of the paper’s content.

**Questions:**

1. Regarding the generation process of pseudo-OOD samples based on ID data, if these generated samples are overly close to the ID distribution, it could lead to confusion in the OOD detector, potentially causing ID samples to be misclassified as pseudo-OOD. Could the authors elaborate on whether any specific techniques were applied during training to control this distributional proximity?
2. Additionally, I noticed that the model exhibits substantial variance in the FPR95 metric on the Amazon and Coauthor datasets. Could the authors clarify whether this variance is linked to the aforementioned distributional control during the generation process?

---

> ### Author Response · Authors · 2024-11-19
> **Response to Reviewer SRXq (1/1)**
>
> We sincerely appreciate your constructive opinions for our paper. We have thoroughly reviewed the concerns and provided the following clarification.
>
> > **Re W1:** Blank Figure
>
> Thank you for pointing out the issue with the possibly missing Figure 1(d), we have attempted to download the manuscript PDF from OpenReview, and the figure appeared to be available. This scatter plot with many dots has used a PNG figure to reduce the file size instead of using a huge PDF file. The rendering issue may potentially be a system error with OpenReview. We have revised the figure (highlighted caption) in the updated manuscript by replacing it with an even smaller-sized PNG image.
>
> > **Re Q1:** Difficulty of pseudo-OOD samples overly close to ID distribution
>
> We thank the reviewer for raising an insightful point about the potential confusion in the OOD detector when pseudo-OOD samples are too close to ID data. In GOLD, pseudo-OOD data is generated via an LGM, which intentionally creates embeddings that initially lie close to ID samples - **this proximity is by design, as it helps create challenging and representative instances**.
>
> To **mitigate any confusion caused by this closeness**, GOLD employs an implicit adversarial optimisation framework along with divergence regularisation (Eq.13), which iteratively maximises the energy gap between ID and pseudo-OOD embeddings. This training strategy ensures sufficient separation between the two distributions while maintaining the challenging nature of the synthetic samples. An illustration of this embedding separation process is provided in the motivation for GOLD in Figure 1 of the manuscript, where the initially close pseudo-OOD samples are effectively separated after training. To further control the proximity and diversity of pseudo-OOD data, GOLD employs different mechanisms based on the generative model used: with the default LDM, diversity is managed through the initial noisy vectors, while in the VAE variant, it is controlled by sampling the mean and variance in the latent space.
>
> Moreover, the **benefit of using difficult pseudo-OOD that are close to ID data** is to prevent the detector from being biased by or overfitting to the OOD data. If the OOD used for training is too easy or too far away from in-distribution data, (e.g., using molecule graph as OOD for social network), the detector will easily overfit towards this direction, which cannot handle the difficult real-world OOD situation that could be much closer to the in-distribution data (e.g., detecting Place A’s data as OOD from Place B’s data in social network).
>
> > **Re Q2:** Large variance in the FPR95 Metric for Amazon and Coauthor dataset
>
> Thank you for your insightful question. The **result in Table 1 is obtained by averaging the different OOD test sets**, where the **variance reflects the difficulty of each of these test sets** instead of the stability of the GOLD model on a single test set. For example, the $3.16 \pm 5.46$ result of GOLD on the Coauthor dataset is obtained from the average performance on three separate test sets with different types of OOD, Coauthor-S (structure manipulation OOD), Coauthor-F (feature interpolation OOD), and Coauthor-L (label leave-out OOD), respectively.
>
> The stability of GOLD is demonstrated by the variance in the detailed evaluation tables for each testing set, which is shown in Table 11~15 in the Appendix. It can be seen that the variance within the individual subsets remains low.
>
> We have revised the clarity in the updated version of the manuscript in the description of Table 1 and Section 4.1 (highlighted).

---

> > ### Comment · Reviewer_SRXq · 2024-11-25
> > **Reply**
> >
> > Thanks to the author for the reply. I have no more questions and will revise my score. Good luck!

---

> ### Author Response · Authors · 2024-11-25
>
> Dear Reviewer SRXq,
>
> We highly appreciate the insightful feedback and the score increase we have received. We will ensure that all the updates are included in the future version of the paper.

---

### Official Review · Reviewer_syps · 2024-11-04

**Soundness:** 3
**Presentation:** 2
**Contribution:** 3
**Rating:** 6
**Confidence:** 2

**Summary:**

The paper presents a framework designed to detect out-of-distribution data without requiring pre-existing OOD datasets or pre-trained models. The proposed GOLD framework includes:
- A latent generative model to generate synthetic embeddings that imitate in-distribution embeddings from a GNN.
- A GNN encoder and OOD detector to classify in-distribution data and maximize energy divergence between in-distribution and synthetic embeddings.

**Strengths:**

- *No OOD Data Required*: Synthesizes pseudo-OOD data through adversarial training, alleviating the need for real OOD samples.
- *Implementation Flexibility*: Supports both LDM and VAE variants, offering trade-offs between performance and computational efficiency.
- *Strong Empirical Performance*: Outperforms non-OOD methods and matches/exceeds methods using real OOD data.

**Weaknesses:**

- Using generative models to generate samples for downstream tasks has been widely adopted in previous methods. For example, generated samples are commonly used in continual learning for experience replay. It seems that the use of generators in GOLD applies the same concept to different tasks. Energy-based detection is also widely used. The major contribution seems to lie in a new divergence regularisation (Eq.12).
- Further ablative studies on divergence regularisations may be needed to better reflect their effectiveness.
- The presentation needs to be improved.

**Questions:**

- Could the author better clarify the key novelty of GOLD compared to existing works? What distinguishes the usage of the generator in GOLD from previous works?
- The training procedure has two stages: one stage involves fixing the GNN and training the LGM, and the second stage involves fixing the LGM and training the GNN. Is it possible to combine these two stages using a gradient reversal layer?
- Further ablative studies on  $L_{DReg}, L_{Unc}, L_{EReg}$  (removing each of them or applying each individually) may help to better demonstrate the effectiveness of the proposed new divergence regularization (Eq. 12).
- What modifications would be needed to extend the framework beyond node-level detection, such as to graph-level OOD detection tasks?

---

> ### Author Response · Authors · 2024-11-19
> **Response to Reviewer syps (1/2)**
>
> We greatly appreciate the time and effort dedicated to providing us with thought-provoking comments and questions. We have thoroughly addressed your concerns.
>
> > **Re W1 & Q1:** Novelty and Contribution of GOLD to existing work
>
> We thank the reviewer for recognising that our proposed **divergence regularisation (Eq.12) is a key novelty**.  Building on this insight, GOLD also introduces a **second key novelty: an implicit adversarial framework** for synthesis-based OOD detection, which differentiates its purpose and realisation from existing works.
>
> **Traditional generative approaches primarily focus on preserving in-distribution (ID) characteristics**, generating samples that retain or mimic information from prior or ID data \[1\]. As the reviewer rightly points out, such methods are widely applied across various tasks, including continual learning for experience replay. When applied specifically to out-of-distribution detection, existing approaches typically use generative models to create samples from low-density boundary regions of ID data, relying on boundary-synthesis approaches based on non-parametric nearest neighbour distances \[2, 3,4\]. These methods create synthetic OOD data by selecting points near the ID boundary, which may result in limited representation of the real OOD space.
>
> **GOLD takes a fundamentally different approach** via our implicit adversarial learning framework. It employs a novel two-phase process: (1) generating ID-like samples using an LGM, and (2) transforming these samples into OOD instances using a novel energy-guided approach within the adversarial framework, guided by OOD-specific regularisation. This iterative process is distinctive both in its objective—producing semantically meaningful pseudo-OOD samples closer to real-world scenarios rather than synthesising boundary samples—and its realisation, as it leverages a unique adversarial optimisation rather than traditional boundary-synthesis techniques.
>
> Thus, the **key novelties of GOLD lie in two interconnected components**: (1) the divergence regularisation for effective OOD detection, and (2) the implicit adversarial framework that iteratively balances ID-like sample generation with purposeful OOD divergence. While generative models and energy-based detection methods are common, GOLD's comprehensive training framework integrates these components in a novel manner, producing pseudo-OOD samples that are more representative and effective compared to prior methods. This framework also offers a practical solution to address the limitation of graph data, where pre-trained generative models like StableDiffusion are unavailable. The effectiveness of GOLD is further validated through empirical studies.
>
> > **Re W2 & Q3:** Ablation studies on different divergence regularisations
>
> Thank you for your suggestion of analysing the regulariser (Reg) ($\mathcal L_{\text{Unc}}$, $\mathcal L_{\text{EReg}}$, and $\mathcal L_{\text{DReg}}$) individually to better illustrate the effectiveness of our proposed method. We present a **detailed evaluation of the various combinations** of these regularisers below (using the Twitch dataset), where a \checkmark indicates the inclusion of a given regulariser (removing or applying each individually):
> |**Twitch**|  $\mathcal{L}_{\text{Unc}}$ | $\mathcal{L}_{\text{EReg}}$ | $\mathcal{L}_{\text{DReg}}$ | AUROC $(\uparrow)$ | AUPR $(\uparrow)$ | FPR95 $(\downarrow)$ | ID Acc |
> |----------|---------|----------|----------|----------|----------|----------|----------|
> |Without Reg|  |  | | 86.44 |80.64 | 79.84 | 68.97 |
> |Applying Reg| &check; |         |         | 10.18 |40.62 | 97.84 | 70.15 |
> |          |         | &check; |         | 78.02 |83.37 | 78.90 | 70.98 |
> |          |         |         | &check; | 69.04 |76.88 | 44.54 | 70.79 |
> |Removing Reg| &check; | &check; |         | 76.88 |81.49 | 76.14 | 70.99 |
> |             | &check; |         | &check; | 64.43 |75.46 | 45.95 | 69.64 |
> |             |         | &check; | &check; | $\color{red}{89.58}$ |$\color{red}{93.12}$ | $\color{red}{43.78}$ | 69.64 |
> |GOLD| &check; | &check; | &check; | $\color{teal}{99.46}$ |$\color{teal}{99.62}$ | $\color{teal}{1.78}$  | 68.49 |
>
> The results demonstrate that the GOLD configuration, which incorporates all three regularisers, achieves the highest performance, highlighting the critical role each regulariser plays in the model’s design.
> Additional **extended analysis on more datasets and subsets** is available in **Table 4 of the main text and Appendix A.9 (Tables 18-20)**.
>
> [1] Goodfellow et.al., Generative Adversarial Networks. NIPS14.
>
> [2] Tao et.al., Non-Parametric Outlier Synthesis. ICLR23.
>
> [3] Du et.al., Dream the Impossible: Outlier Imagination with Diffusion Models. NIPS23.
>
> [4] Lee et.al., Training Confidence-calibrated Classifiers for Detecting Out-of-Distribution Samples. ICLR18.
>
> [5] Liu et.al., Towards Graph Foundation Models: A Survey and Beyond. 2024.

---

> > ### Author Response · Authors · 2024-11-19
> > **Response to Reviewer syps (2/2)**
> >
> > > **Re Q2:** Combining the two-step training procedure via gradient reversal layer
> >
> > We would like to thank the reviewer for the excellent idea, gradient reversal layer (GRL) has demonstrated great success in domain adaption \[1\], and we hope to study the use of GRL as a future work. We believe it is possible to combine the two training stages using a GRL, enabling an end-to-end schema that optimises both the latent generative model, classifier and detector simultaneously. This approach has several advantages, including simplifying the training pipeline by unifying the objectives and potentially reducing overall training time. Nonetheless, GRL raises potential stability concerns, as the classifier/detector may converge too quickly early in training, causing the gradient to vanish \[2\]. This contrasts with the two-stage training approach, where independent objectives for the generator, classifier and detector provide stronger gradients to the target mapping \[2, 3\]. The original two-stage method, though requiring separate training phases, potentially ensures a more robust gradient flow through its independent optimisation objectives. Nevertheless, we aim to investigate the adversarial generation behaviour by utilising GRL in GOLD in future work, given the various potentials in further improving the training paradigm.
> >
> > > **Re Q4:** Extension to graph-level OOD detection
> >
> > Thank you for your insightful question and we agree with you that there will be of great significance regarding the extension of our node-level OOD detection method to graph-level OOD detection. As mentioned in the Potential Limitation section in Appendix A.4, our work focuses on node-level OOD detection, which aims to identify OOD nodes within a graph. This approach emphasises localised graph structures and the influence of neighbouring nodes. In contrast, graph-level OOD detection involves identifying whether an entire graph is out-of-distribution, based on the global properties of in-distribution graphs. The distinction between these tasks requires different model architectures and evaluation strategies.
> >
> > While our current framework is tailored for node-level tasks, the underlying principles (e.g., leveraging adversarially generated samples to improve OOD detection) could be extended to graph-level OOD detection. For example, instead of generating in-distribution-like nodes, we could generate in-distribution-like graphs using the proposed generative methods. The discrimination mechanism (implicit adversarial training) could then be adapted to evaluate entire graphs based on the learned graph representations.
> >
> > We appreciate your suggestion and recognise that extending the proposed approach to graph-level OOD detection is a promising direction. We aim to explore this adaptation in future work.
> >
> > > **Re W3:** Improved presentation
> >
> > We would like to extend our gratitude for your constructive feedback, and we kindly ask for some details on how to improve the presentation of our manuscript. We believe your expertise will greatly enhance the quality of our work.
> >
> > [1] Ganin et.al., Unsupervised Domain Adaptation by Backpropagation. ICML15.
> >
> > [2] Tzeng et.al., Adversarial Discriminative Domain Adaptation. CVPR17.
> >
> > [3] Bhattacharya et.al., Generative Adversarial Speaker Embedding Networks for Domain Robust End-to-End Speaker Verification. ICASSP19.

---

> > > ### Comment · Reviewer_syps · 2024-11-25
> > >
> > > Dear authors,
> > >
> > > Thanks for your reply.
> > >
> > > Now I understand the novelty of this paper better. From the ablation studies, it seems that $L_{EReg}$ plays a more significant role in performance than $L_{DReg}$. However, it is good to see that the combined loss achieves the optimal performance.
> > >
> > > I think my concerns are addressed. I will keep my scores.

---

> > > > ### Author Response · Authors · 2024-11-25
> > > >
> > > > We sincerely thank you for your constructive feedback and for recognising the strengths of our work.
> > > >
> > > > We also appreciate your careful attention to detail and acknowledge the effectiveness of our combined loss. To kindly highlight, our proposed **$\mathcal L_{\text{DReg}}$ plays a crucial role beyond $\mathcal L_{\text{EReg}}$**, significantly reducing the FPR95 (lower is better) from 78.90% to 44.54% compared to using $\mathcal L_{\text{EReg}}$, when they are both used alone. Here, the FPR95 focuses on performance under high-sensitivity conditions—specifically, the rate of misclassifying ID samples as OOD when the true positive rate is at 95%. Thus, **FPR95 provides a more noticeable score gain** under stricter criteria, reflecting greater improvements in detection performance.
> > > >
> > > > As highlighted in the ablation table, **$\mathcal L_{\text{DReg}}$ markedly lowers the FPR95** but comes with a trade-off in AUROC (higher is better). In contrast, **$\mathcal L_{\text{EReg}}$ ensures a higher AUROC** but results in a much larger FPR95. The combination of these regularisers achieves the desired balance, providing both a high AUROC and a low FPR95, which are critical for OOD detection (using only $\mathcal L_{\text{DReg}}$ and $\mathcal L_{\text{EReg}}$)  . Furthermore, incorporating $\mathcal L_{\text{Unc}}$ ensures the detector produces more distinct outputs for ID and OOD samples, leading to a significant improvement in OOD detection performance (e.g., FPR95: 1.78%).
> > > >
> > > > Apologies for the potential ambiguity in the last version of the table and we have updated the ablation table to better reflect the interpretations of the metrics.
> > > >
> > > > Thank you again for your insightful reply!

---

### Author Response · Authors · 2024-11-29
**Global Summary**

Dear Program Chairs, Senior Area Chairs, Area Chairs, and Reviewers,

We sincerely appreciate the **insightful and thought-provoking feedback** provided by the reviewers, which has been invaluable in improving our manuscript. Below, we kindly **summarise the clarifications and additional experimental results** we have included to emphasise the novelty and effectiveness of our proposed method, GOLD:

- **Novelty and Dataset Analysis:** We have added a detailed discussion on the novelty of our proposed method, GOLD, and provided an analysis of its performance achievements in relation to the dataset characteristics, addressing the questions raised by Reviewers [syps](https://openreview.net/forum?id=y5einmJ0Yx&noteId=pXg8jFFWO0) and [yx8w](https://openreview.net/forum?id=y5einmJ0Yx&noteId=2GoYn0qWiQ).

- **Pseudo-OOD Generation and FPR95 Variance:** In response to Reviewer [SRXq](https://openreview.net/forum?id=y5einmJ0Yx&noteId=sUpGv5ZNwA)’s query, we clarified the process of generating pseudo-OOD data and enhanced the explanation of the variance in FPR95 results for improved clarity.

- **Backbones and Regularisation Losses:** We conducted additional experiments on various backbones and regularisation losses as recommended by Reviewers [AALT](https://openreview.net/forum?id=y5einmJ0Yx&noteId=opxXrudfo3) and [syps](https://openreview.net/forum?id=y5einmJ0Yx&noteId=pXg8jFFWO0).

- **Technical Details and Future Work:** We provided further clarification on technical details in response to each reviewer's questions and expanded the discussion to potential directions for future work.

We are deeply grateful for the constructive feedback, which has significantly strengthened our work. All extended clarifications and results will be incorporated into the updated version of our paper.

Best regards,

Authors 2927

---

### Meta-Review · Area_Chair_WtsB · 2024-12-08

**Metareview:**

This paper introduces GOLD, a novel framework addressing out-of-distribution (OOD) detection challenges in graph neural networks (GNNs). By leveraging an implicit adversarial training pipeline, the framework generates pseudo-OOD embeddings without relying on pre-trained generative models or additional OOD datasets. The core innovation lies in its alternating optimization framework, which effectively balances in-distribution (ID) representation and divergence-enhanced pseudo-OOD generation. Comprehensive experiments across five benchmark datasets demonstrate the superior performance of GOLD over state-of-the-art baselines, with significant gains in metrics like FPR95 and AUROC.

Strengths:
* Novelty: GOLD’s use of implicit adversarial learning for OOD detection is novel and effectively addresses limitations in current methods.
* Practicality: Eliminating the need for pre-trained generative models or auxiliary OOD datasets increases the framework’s applicability across diverse graph data scenarios.
* Empirical Evidence: Extensive experiments across multiple datasets confirm the robustness of the method, with notable improvements in FPR95 on challenging datasets like Twitch.
* Clarity: The manuscript is well-structured, and responses to reviewers adequately clarify technical details, including performance variance and metric analysis.

Weaknesses and Revisions:
* Presentation: Initial clarity issues (e.g., ambiguous notation, a rendering issue in Figure 1(d)) were resolved in the revised manuscript.
* Ablation Studies: Reviewers requested more detailed analysis of regularizers and training techniques, which the authors provided, showing the necessity of all components for optimal performance.
* Computational Cost: While acknowledged as a limitation, the trade-off for improved performance is justified and partially mitigated with alternative backbones like VAE.

Overall, this paper represents a meaningful contribution to OOD detection in graph-structured data. Its methodological novelty, empirical rigor, and practical implications outweigh minor presentation issues, most of which were addressed during the review process. The reviewers consistently rated the work above the acceptance threshold, therefore an accept decision is recommended.

**Additional Comments On Reviewer Discussion:**

Points Raised by Reviewers:

syps:
* Clarification on GOLD's novelty compared to existing methods.
* Suggestion for combining training stages with a gradient reversal layer.
* Request for more ablation studies on divergence regularizations.
* Inquiry on extending the method to graph-level OOD detection.
* Need for improved presentation.

SRXq:
* Query on controlling distributional proximity of pseudo-OOD samples.
* Clarification needed on variance in FPR95 metric across datasets.

AALT:
* Explanation of notation in equation (10).
* Inquiry about experiments with different backbone models.

yx8w:
* Request for deeper analysis of performance on FPR95 dataset.

Author Responses:

syps:
* Clarified the novelty of GOLD, especially the divergence regularization and adversarial framework.
* Acknowledged potential for using gradient reversal layers in future work.
* Provided ablation studies on divergence regularizations.
* Discussed extending to graph-level OOD detection as future work.
* Improved presentation based on feedback.

SRXq:
* Explained the control of distributional proximity in pseudo-OOD generation through adversarial training.
* Clarified variance in FPR95 by detailing the methodology of dataset evaluation.

AALT:
* Clarified the notation in equation (10).
* Conducted and reported experiments with GAT and MixHop backbones.

yx8w:
* Provided an in-depth analysis of FPR95 performance, correlating with dataset characteristics.

---

### Decision · Program_Chairs · 2025-01-22

Accept (Spotlight)